# Using atmospheric trace gas vertical profiles to evaluate model fluxes: a case-study of Arctic-CAP observations and GEOS simulations for the ABoVE domain

Colm Sweeney[1], Abhishek Chatterjee[2,3], Sonja Wolter[4,1], Kathryn McKain[4,1], Robert Bogue[5*], Tim Newberger[4,1], Lei Hu[4,1], Lesley Ott[3], Benjamin Poulter[3], Luke Schiferl[6], Brad Weir[2,3], Zhen Zhang[7], Charles E. Miller[5]

[1]NOAA Earth System Research Laboratory, Boulder, CO, USA
[2]Universities Space Research Association, Columbia, MD, USA
[3]NASA Goddard Space Flight Center, Greenbelt MD, USA
[4]CIRES, University of Colorado, Boulder, CO, USA
[5]Jet Propulsion Laboratory, California Institute of Technology, Pasadena CA, USA
[6] LDEO, Columbia University, New York, NY, USA
[7]University of Maryland, College Park, MD, USA
 * now at McGill University, Montreal, QC, Canada

*Correspondence to*: Colm Sweeney (colm.sweeney@noaa.gov)

**Abstract.** Accurate estimates of carbon-climate feedbacks require an independent means for evaluating surface flux models at regional scales. The altitude-integrated enhancement (AIE) derived from the Arctic Carbon Atmospheric Profiles (Arctic-CAP) project demonstrate the utility of this bulk quantity for surface flux model evaluation. This bulk quantity leverages background mole fraction values from the middle free troposphere, are agnostic to uncertainties in boundary layer height, and can be derived from model estimates of mole fractions and vertical gradients. To demonstrate the utility of the bulk quantity, six airborne profiling surveys of atmospheric carbon dioxide ($CO_2$), methane ($CH_4$) and carbon monoxide (CO) throughout Alaska and northwestern Canada between April and November 2017 were completed as part of NASA's Arctic-Boreal Vulnerability Experiment (ABoVE). The Arctic-CAP sampling strategy involved acquiring vertical profiles of $CO_2$, $CH_4$ and CO from the surface to 5 km altitude at 25 sites around the ABoVE domain on a 4- to 6-week time interval. All Arctic-CAP measurements were compared to a global simulation using the Goddard Earth Observing System (GEOS) modeling system. Comparisons of the AIE bulk quantity from aircraft observations and GEOS simulations of atmospheric $CO_2$, $CH_4$ and CO highlight the fidelity of the modeled surface fluxes. The model-data comparison over the ABoVE domain reveals that while current state-of-the-art models and flux estimates are able to capture broadscale spatial and temporal patterns in near-surface $CO_2$ and $CH_4$ concentrations, more work is needed to resolve fine-scale flux features that are captured in CO observations.

## 1. Introduction

There are many uncertainties to predicting the impact of increased emissions of $CO_2$ and $CH_4$ in the atmosphere. Carbon-climate feedbacks (Arora et al., 2020) are among the most uncertain climate feedbacks. Without a better understanding of how changes in temperature, $CO_2$ itself, water and

nutrients are magnifying or reducing the impact of increased emissions of greenhouse gases (GHG), it will be difficult to use climate models to accurately predict climate change. This uncertainty not only stems from a poor mechanistic understanding of how the biosphere will respond at the smallest scales but also how changes in the landscape drive changes in local environments.

The Arctic, in particular, is a region where carbon-climate feedbacks are critical to understand given the vast quantities of carbon sequestered in the permafrost soils of the northern high latitudes (Hugelius et al., 2014). Rapid changes in temperature have led to concerns about the potential for significant carbon emissions due to changes in ecosystems, permafrost and large-scale disturbances like fires (Schuur et al., 2015; McGuire et al., 2018; Turetsky et al., 2020). Our understanding of the magnitude and behavior of the carbon system response to these changes is rudimentary (Koven et al., 2015). For instance, release of carbon from the permafrost pool could result in increased emissions of $CH_4$ from anaerobic degradation; increased emissions of $CO_2$ from aerobic degradation; increased uptake of carbon due to new availability of nutrients and above-ground ecosystem growth; or an increase in mobilization of carbon through runoff. Alternatively, increases in disturbances such as fires may significantly impact below-ground carbon storage, uptake of $CO_2$ and emissions of $CH_4$, CO, and $CO_2$. Limitations in our understanding of the accuracy of modeled fluxes of $CO_2$, CO and $CH_4$ have resulted in large uncertainties in the magnitude of Arctic carbon-climate feedbacks (e.g., Koven et al., 2011; Schneider von Deimling et al., 2012; Schaefer et al., 2014; Lawrence et al., 2015; Schuur et al., 2015). The lack of observations from which to build and evaluate models of the biosphere is a significant source of the problem and leads to both enhanced uncertainty and reduced fidelity in our model simulations. In general, land- and ocean-atmosphere fluxes from climate models are most commonly evaluated using flux measurements made with eddy covariance or flux chamber techniques (Sasai et al., 2007). While flux measurements of these types are widely available over many ecosystem types, they represent the impact of limited spatial domains that are rarely more than a 1000 m radius around a given site (Schmid, 2002; Gockede et al., 2005) and may be significantly smaller depending on topography, wind direction, boundary layer stability, and measurement approach. Land surface inhomogeneities within these small footprints (Baldocchi et al., 2005) and regional-scale (100-1000 km scales) variability of these ecosystems can lead to significant biases when eddy covariance measurements are scaled up to represent large areas (e.g. Mekonnen et al., 2016). This is especially true in the Arctic where microtopography can result in fluxes varying by orders of magnitude on a scale of 1-100 meters (Johnston et al., 2014).

An alternative to the "bottom-up" evaluation approach, which relies on the eddy covariance measurements, is the "top-down" approach, which makes use of atmospheric measurements of species like $CO_2$, $CH_4$ and CO and modeled atmospheric transport patterns to infer the surface fluxes needed to reproduce observed atmospheric concentrations (Pickett-Heaps et al., 2011; Miller et al., 2016; Thompson et al., 2017 are examples in the Arctic) over large regional scales. In a data limited region, this inverse approach generally takes a forward-flux model, or a set of observations that are likely correlated with the flux, as a prior or first guess. The inverse approach then estimates the flux by scaling the prior. While the inverse approach results in a flux estimate that meets the constraint of the trace gas measurements and modeled transport, the variability in surface flux from these analyses cannot be directly attributed to mechanisms such as temperature changes, $CO_2$ fertilization, nutrient enrichment and water stress and, therefore do not have any predictive capabilities. Also, inverse methods are

influenced by errors in atmospheric transport and assumptions about error covariances, which are difficult to characterize (Gourdji et al., 2012; Lauvaux et al., 2012; Mueller et al., 2018; Chatterjee and Michalak, 2013).

In this study, a hybrid approach is taken to evaluate and benchmark the accuracy of current state-of-the-art bottom-up land-surface flux models using a bulk quantity calculated from atmospheric vertical profiles of trace gas mole fractions. The goal is to present an approach to evaluate land-surface flux models that capture complex carbon cycle dynamics over the northern high-latitudes. NASA's Goddard Earth Observing System (GEOS) general circulation model (GCM) is used with a combination of surface flux components for $CO_2$, $CH_4$ and CO to create 4D atmospheric fields; these fields are subsequently evaluated using the altitude-integrated enhancements (AIE) calculated from profiles collected during the Arctic Carbon Atmospheric Profiles (Arctic-CAP) airborne campaign.

Both the Arctic-CAP project and the GEOS model runs for the domain are part of NASA's Arctic Boreal Vulnerability Experiment (ABoVE, www.above.nasa.gov), a decade-long research program focused on evaluating the vulnerability and resiliency of the Arctic tundra and boreal ecosystems in western North America (Miller et al., 2019). One of the primary objectives of the ABoVE program is to better understand the major processes driving observed trends in Arctic carbon cycle dynamics, in order to understand how the ecosystem is responding to environmental changes and to characterize the impact of climate feedbacks on greenhouse gas emissions. ABoVE has taken two approaches to better understand critical ecosystem processes vulnerable to change. The first is through ground-based surveys and monitoring sites in representative regions of the ABoVE domain. These multi-year studies provide a backbone for intensive investigations, such as airborne deployments. The Arctic-CAP campaign discussed here was one such airborne deployment that was conducted during the spring-summer-fall of 2017 (Section 2.1). The subsequent analysis described here illustrates how improvements in surface models develop through ground-based surveys, and monitoring sites can be evaluated and tested over larger spatial scales using aircraft profiles (Section 3). This study uses the bulk quantify from Arctic-CAP aircraft profiles to directly evaluate the terrestrial surface flux models of $CO_2$, $CH_4$ and CO. For the sake of demonstration, we rely on one transport model and one flux scenario for each tracer (i.e., $CO_2$, $CH_4$ and CO) to show the utility of the three carbon species to diagnose and identify deficiencies in the land flux models. Ongoing and future studies will build upon the results discussed here and further diagnose transport and flux patterns from multiple models based on additional aircraft and ground-based observations throughout the ABoVE domain. This approach demonstrates the value of aircraft profiles.

## 2. Methods

### 2.1. Arctic-CAP Flight Planning and Sampling Strategy

Arctic-CAP was designed to measure vertical profiles of atmospheric $CO_2$, $CH_4$ and CO mole fraction to capture the spatial and temporal variability of carbon cycle dynamics (Sweeney et al., 2015; Parazoo et al., 2016) across the ABoVE domain. Six campaigns were performed during 2017: late April – early May, June, July, August, September, and late-October – early November. Arctic-CAP flights surveyed the ABoVE Study Area and were organized around an Alaskan circuit and a Canadian circuit (Fig. 1).

The Alaskan circuit covered a region where aircraft measurements were previously made during 2012-
2015 by the Carbon in Arctic Reservoirs Vulnerability Experiment (CARVE; Miller et al., 2012), which
included the Alaskan Boreal Interior, Brooks Range Tundra and the Alaskan Tundra ecoregions. The
Arctic-CAP Alaska circuit was primarily west of Fairbanks, Alaska, and include Galena, Bethel,
Unalakleet, Nome and Kotzebue. The northern section of the circuit overflew Utqiagvik (formerly
Barrow), Atqasuk, Deadhorse and the Toolik Lake Research Station – all North Slope tundra sites with
long-term measurements of atmospheric $CO_2$ and $CH_4$. The Arctic-CAP Canadian circuit focused on
flying over sites in and around the Inuvik and Yellowknife areas in the Canadian Arctic. In the Inuvik
region, the aircraft overflew the Trail Valley Creek and Havipak Creek research sites, and the Daring
Lake and Scotty Creek flux tower sites were overflown on the way to and from the Yellowknife area.
The Canadian Circuit expands upon the ecoregions covered in the CARVE missions to include the
Boreal Cordillera, Taiga Plain, Taiga Shield and the Southern Arctic Tundra ecoregions.
Approximately 25 vertical profiles were acquired during each campaign (Fig 2). The majority of each
flight day was spent in the well-mixed boundary layer with 2-4 vertical profiles up to altitudes of 5000
m above sea level (masl). Using missed approaches to get as near to the ground as possible, profiles
diagnosed temporal changes in the boundary layer and residual layers above where surface fluxes may
have recently (< 3 days) influenced that atmospheric column. During the 2017 season, Arctic-CAP
flights were complemented by additional vertical profiles collected in the ABoVE domain by the
ASCENDS (Active Sensing of $CO_2$ Emissions over Nights, Days, & Seasons, https://www-
air.larc.nasa.gov/cgi-bin/ArcView/ascends.2017?MERGE=1) and ATom (Atmospheric Tomography,
Wofsy et al., 2018) campaigns and the NOAA Carbon Cycle Aircraft Program (Karion et al., 2015;
Sweeney et al., 2015). The focus of this study will be on the $CO_2$, $CH_4$ and CO data acquired during
Arctic-CAP and, in particular, utilizing the profiles acquired during each flight to separate signals from
near field surface fluxes from large-scale deviations in a way that is agnostic to model errors due to
inaccurate vertical transport.

## 2.2.    Aircraft and Payload

Arctic-CAP flights were performed with a Mooney Ovation 3 aircraft (tail number N617DH, Scientific
Aviation). The Mooney operated at a cruise speed of 170 kts and reached profile altitudes of 5 km
(17,000 feet) on each flight, with most legs lasting 4-5 hours and covering an average distance of ~1350
km. The average ascent and descent rates were limited to ~100 m/min to minimize hysteresis in the
temperature and relative humidity measurements. The basic research payload flown on all six research
missions included continuous in-situ $CO_2$, $CH_4$, CO, $H_2O$, temperature and horizontal winds. The in-situ
measurements (Sweeney and McKain, 2019) followed the methodology described in Karion et al.
(2013), and wind measurements followed the protocol outlined in Conley et al. (2014). During Arctic-
CAP, insitu measurements of $CO_2$, $CH_4$ and CO were made every ~2.4 s and aggregated to 10 s
averages for comparison the GEOS 4D fields (latitude, longitude, altitude and time). Sampling at the 10
s resolution reduces the spatial representativeness error between the model grid cell and the aircraft
observations.
Programmable flask packages (PFPs; Sweeney et al., 2015) provided an independent check of the
calibration scale of the continuous *in situ* $CO_2$, $CH_4$ and CO measurements, as well as samples for more

than 50 different species including $N_2O$, $SF_6$, and a variety of hydrocarbons, halocarbons and isotopes of carbon (Sweeney et al., 2020). Carbonyl sulfide measured in the flask samples can be used as a tracer of gross primary productivity (GPP) (Montzka et al., 2007), while ethane, propane and C-13 isotope of $CH_4$ provide another constraint on the source of the $CH_4$ emissions. Each flight sampled a single 12-flask package providing a total of ~84 flasks per research mission to better understand the factors

controlling local fluxes of $CO_2$, $CH_4$ and CO and the long-range transport of these species from low latitudes.

### 2.3. GEOS Earth System Model & Atmospheric $CO_2$, CO and $CH_4$ Modelling

The GEOS (Rienecker et al., 2011; Molod et al., 2015) model is a complex yet flexible modeling system that describes the behavior of the land and atmosphere on a variety of spatial (~12.5-100 km)

and temporal (hourly to decadal) scales. GEOS includes both an atmospheric General Circulation Model (GCM) and data assimilation system that have been used to produce the widely-used Modern-Era Retrospective Analysis for Research and Applications (MERRA) (Rienecker et al., 2011) and MERRA-2 (Bosilovich et al., 2015; Gelaro et al., 2017). The GEOS Forward Processing (GEOS FP) system produces atmospheric analyses and 10-day forecasts in near real-time, which are used to provide

forecasting support to NASA field campaigns and satellite instrument teams (e.g. Strode et al., 2018). GEOS has also been used extensively to study atmospheric carbon species (e.g. Allen et al., 2012; Ott et al., 2015; Weir et al., 2021).
The GEOS setup utilized in this work simulates $CO_2$, CO and $CH_4$ simultaneously at nominal 0.5° horizontal resolution, 72 vertical layers (up to ~0.1 hPa) with trace gas output saved every 3-hours. For

$CO_2$, the surface fluxes consist of 5 different components from a Low-order Flux Inversion (LoFI) package (Weir et al., 2021): 1) net ecosystem exchange (NEE) from the Carnegie Ames Stanford Approach - Global Fire Emissions Database (CASA-GFED) mode with a parametric adjustment applied to match the atmospheric growth rate (Weir et al., 2021), 2) anthropogenic biofuel burning emissions, i.e., harvested wood product (Van Der Werf et al., 2003), 3) biomass burning emissions derived from

the fire radiative power based Quick Fire Emissions Dataset (QFED; Darmenov and Da Silva, 2015) 4) fossil fuel emissions from the Open-source Data Inventory for Anthropogenic $CO_2$ (ODIAC; Oda and Maksyutov, 2011), and 5) ocean exchange fluxes based on *in situ* measurements of the partial pressure of $CO_2$ in sea-water from the Takahashi et al. (2009) dataset but adding back the inter-annual variability and applying a mean ocean partial pressure of $CO_2$ growth rate of 1.5 $\mu$atm/yr at each point every year.

For CO, the emissions include biomass burning emissions from QFED, and climatologies of fossil fuel and biofuel emissions and VOC fields (Duncan et al., 2007; Ott et al., 2010). Finally, the $CH_4$ flux collection consists of five components: 1) wetland emissions from the process-based ecosystem model LPJ-*wsl* (Lund-Potsdam-Jena model, WSL version - Poulter et al., 2011), 2) biomass burning emissions from the QFED, 3) industrial and fossil fuel emissions from the Emissions Database for Global

Atmospheric Research (EDGAR v4.3.2, Janssens-Maenhout et al., 2017; Crippa et al., 2018), 4) agricultural emissions from EDGAR v4.3.2 and 5) anthropogenic biofuel burning emissions from EDGAR v4.3.2. Note that since the EDGAR v4.3.2 emissions record ends in 2012, the same set of values from 2012 were used for the year 2017. As shown later, this is not a bad assumption considering that for the majority of the ABoVE domain, the most critical $CH_4$ emissions are from the wetlands

sector. On the other hand, care was taken to use a version of the LPJ-*wsl* model that includes a state-of-

the-art hydrology subroutine (TOPMODEL) to determine wetland area and its inter- and intra-annual dynamics (Zhang et al., 2016), a permafrost and dynamic snow model (Wania et al., 2009) with explicit representation of the effects of snow and freeze/thaw cycles on soil temperature and moisture, and thus the $CH_4$ emissions. Table 1 provides a summary of the flux components, their specifications and
associated references.

## 2.4.    AIE calculation

As will be explained in the following results section the surface fluxes of $CO_2$, $CH_4$ and CO in GEOS are compared to aircraft observations by first subtracting the average daily free tropospheric value
(>3000 m for $CO_2$ and $CH_4$ and >4000 m for CO, $X_{FT}$) from each measurement below 3000 m and comparing that to the altitude integrated sum

$$\Delta X = \int_{z=ground}^{z=3000} ((X - X_{FT})/n_{BL})\, ndz \qquad\qquad \text{Eq. 1}$$

where $\Delta X$ is altitude-integrated sum of the mole fraction of species X minus $X_{FT}$ divided by the $n_{BL}$ where $n_{BL} = \int_{z=ground}^{z=3000} n\, dz$ and n is the atmospheric number density. It is assumed that the mole
fraction of each trace gas species measured at the lowest point in each profile is constant to the ground level. Ground level altitude is taken from USGS (USGS, 2017). Thus, the AIE is equivalent to average enhancement in the boundary layer after accounting for altitude changes in number density. As will be explained in the results section, the 3000 m was picked as cutoff for $CO_2$ and $CH_4$ because of the low variability of these tracers above that altitude level where as the cutoff point for CO was chosen to be
4000 m.

**Table 1. Components of fluxes for simulation of atmospheric concentrations of $CO_2$, CO and $CH_4$ in GEOS. Flux components that are the primary drivers of observed signals within our study domain are distinguished with italics.**

| Flux type | Used in simulation of | Inventory / Process-based model name | Reference |
|---|---|---|---|
| Fossil fuel | $CO_2$ | ODIAC | Oda and Maksyutov, 2011 |
| Biofuel | $CO_2$ | CASA-GFED3 | Van Der Werf et al., 2003 |
| *NEE* | $CO_2$ | LoFI CASA | Weir et al., 2021 |
| Ocean | $CO_2$ | LoFI Takahashi | Weir et al., 2021 |
| Biomass Burning / Fires | $CO_2$, CO, | QFED | Darmenov and Da Silva, 2015 |
| Fossil fuels & biofuels | CO | Climatology | Duncan et al., 2007 |
| VOC | CO | GMI climatology | Duncan et al., 2007 |
| *Wetlands* | $CH_4$ | LPJ-wsl | Poulter et al., 2011, Zhang et al., |
| Agriculture and waste | $CH_4$ | EDGAR v4.3.2 | Crippa et al., 2018 |
| Biofuels | $CH_4$ | EDGAR v4.3.2 | Crippa et al., 2018 |
| Industrial and fossil fuel | $CH_4$ | EDGAR v4.3.2 | Crippa et al., 2018 |

**Table 2. GEOS $CO_2$ Flux Estimates (PgC yr-1) for 2017. Flux emissions are specified for (a) the natural land sink component, which includes the sum of NEE and biomass burning, and (b) all anthropogenic source components, which include fossil fuel and biofuel burning.**

| ABoVE Domain | | pan-Arctic (>48 N) | | Global | |
|---|---|---|---|---|---|
| Land Sink | Fuel Sources | Land Sink | Fuel Sources | Land Sink | Fuel Sources |
| -0.32 | 0.11 | -1.84 | 1.37 | -3.28 | 11.08 |

We have assessed global and pan-Arctic budgets and compared against existing studies (Tables 2 and 3)
and estimates to establish the fidelity of the model fluxes for large-scale assessments. $CO_2$ flux
estimates indicate that the ABoVE domain is a 0.32 PgC sink for our study year, 2017. This represents
~17% of the calculated pan-Arctic terrestrial carbon sink, which is consistent with the fraction of the
land area > 48N represented by the ABoVE domain (~16%). Perhaps more significantly, the 1.84 PgC
pan-Arctic sink represents 56% of the global sink for 2017. We attribute this large uptake to the vast
boreal forests > 48 N, particularly in Siberia (Sasakawa et al., 2013), where the contemporary Arctic
tundra is thought to be nearly carbon neutral with uncertainties allowing for a small to moderate sink or
a small source (McGuire et al., 2016). These findings are also consistent with Wunch et al. (2013) who
used GOSAT satellite data and TCCON ground-based column measurements to determine that
interannual variability in Northern Hemisphere $CO_2$ uptake was dominated by changes in the boreal
forest. More recent studies, such as Welp et al. (2016) and Commane et al. (2017) have also used
atmospheric inversions to highlight that >90% of the carbon sink in the northern high latitudes reside in
the boreal forests. Our simple forward model simulations and the Arctic-CAP data provide a unique
opportunity to assess the validity of these previous findings over the ABoVE domain. Sub-regional flux
estimates within the ABoVE domain are part of ongoing investigations and will be captured in future
studies.

**Table 3. GEOS CH$_4$ Flux Estimates (TgCH$_4$ yr-1) for 2017. CH$_4$ flux emissions are specified for (a) the wetland component, and (b) all source components, which include wetlands, industrial and fossil fuel, agriculture and waste, biomass burning, biofuel burning and other natural emissions.**

| ABoVE Domain | | pan-Arctic (>48 N) | | Global | |
|---|---|---|---|---|---|
| Wetland | All Sources | Wetland | All Sources | Wetland | All Sources |
| 9.01 | 11.64 | 21.74 | 52.03 | 187.39 | 536.01 |

Examination of the specified CH$_4$ flux estimates for the ABoVE domain (Table 4) reveal a remarkable result: 78% of the emissions, 9.01 TgCH$_4$ yr$^{-1}$, come from wetlands. Furthermore, ABoVE wetlands emissions account for 41% of pan-Arctic CH$_4$ wetland emissions. Both results suggest a disproportionately large contribution of North American wetlands to the regional CH$_4$ budget. Placing this in a larger context, the 52 TgCH$_4$ yr$^{-1}$ from all pan-Arctic emissions account for only about 10% of the global emissions. Our pan-Arctic CH$_4$ emissions estimate of 52 TgCH$_4$ yr$^{-1}$ is only 60% of the 82-84 TgCH$_4$ yr$^{-1}$ determined by Thompson et al. (2017) for latitudes > 50N and the period 2005-2013. The reasons for this large discrepancy are unclear, particularly since the Thompson et al. (2017) study derived their estimate from an inversion of atmospheric CH$_4$ observations; previously, such top-down estimates have tended to be lower than most forward model emissions estimates. Subtracting the 11 TgCH$_4$ yr$^{-1}$ we estimate for the ABoVE domain from our pan-Arctic value leaves 41 TgCH$_4$ yr$^{-1}$ for the remainder of the pan-Arctic. Future work with additional observations and model simulations will help us understand how specific sectors in the ABoVE domain can better capture the complexity of pan-Arctic CH$_4$ emissions. Our overall model value of 536 TgCH$_4$ yr$^{-1}$ for global CH$_4$ emissions in 2017 falls just outside the range of annual emissions estimates for the decade 2008-2017 (Saunois et al., 2019). This discrepancy is primarily due to the fact that we are looking at different time periods and, unlike Saunois et al. (2019), we do not extrapolate the EDGARv4.3.2 dataset using the extended FAO-CH$_4$ emissions and/or British Petroleum statistical review of fossil fuel production and consumption (see Equation 1 in Saunois et al., 2019); instead, we adopt a much simpler approach of repeating the EDGARv4.3.2 from 2012 for the year 2017. Contrary to the emissions from the coal, oil and gas sector, our wetland methane flux emissions are obtained from the LPJ-*wsl* model (Table 1). LPJ-*wsl* is one of the prognostic models that provide wetland emission estimates to the global methane budget (Table 2 in Saunois et al., 2019). It is not surprising then that our global wetland CH$_4$ emission estimates for 2017 is in line with both the bottom-up (100-183 TgCH$_4$ yr$^{-1}$) and top-down (155-217 Tg CH$_4$ yr$^{-1}$) estimates used in the global methane budget estimate.

## 3. Results and Discussion

### 3.1. Analysis of Profiles

**Table 4. Arctic-CAP 2017 campaign summary**

| Campaign | Start (DOY) | End (DOY) |
|---|---|---|
| Apr/May | 116 | 124 |
| June | 157 | 170 |

| | | |
|---|---|---|
| July | 190 | 202 |
| August | 229 | 242 |
| September | 251 | 271 |
| Oct/Nov | 291 | 310 |

Vertical profiles of $CO_2$, $CH_4$ and CO were acquired during 56 flights over the six Arctic-CAP campaigns from late April (day of year (DOY) 116) through early November (DOY 310) 2017 (Table 4). Figure 4 presents the composite vertical profile data for each campaign. The monthly composite
$CO_2$, $CH_4$ and CO vertical profiles capture the expected variations in the seasonal cycle. The composite profiles also show more variability in the boundary layer (altitudes < 3000 masl) within each month and across months than in the free troposphere for $CO_2$ and $CH_4$ (altitudes > 3000 masl). Unlike $CO_2$ and $CH_4$, CO variability in the free troposphere is significantly greater in July and October than the boundary layer showing either long-range transport of CO or CO injected high (>3000 masl) into the
troposphere by local wildfires.

A clearer picture of the vertical gradients between the free troposphere and the boundary layer can be seen by subtracting free tropospheric means from measurements below 3000 masl. The $CO_2$ gradients between the measurements below 3000 masl and average daily free troposphere values show a drawdown in the boundary layer for most of the profiles starting in June and lasting until the end of the
September campaign (Fig. 5). The drawdown signal in $CO_2$ over the Northern Alaska Tundra (often referred to as the "North Slope") was most pronounced in mid-July and continued through the September campaign. The $CO_2$ drawdown in the more southerly regions of the Boreal Cordillera and Alaskan Boreal Interior peaked in August. By the October campaign many regions were showing significant enhancements in the boundary layer $CO_2$ mole fraction relative to the free troposphere. On
the other hand, for both $CH_4$ and CO, significant enhancements were observed from June through early November. Methane enhancements over the Northern Alaska Tundra were observed from July onward, consistent with patterns observed at the long-term surface monitoring station in Utqiaġvik (Sweeney et al., 2016). Similarly, boundary layer $CO_2$ and $CH_4$ are both most enhanced in September and October on the North Alaska Tundra. Due to the high variability in CO above 3000 masl during July and
October (Fig. 4), it is more difficult to use this approach to derive CO enhancements from surface fluxes. To avoid the impact of fire-based CO that has been injected into the free troposphere, the mean background value is taken from measurements above 4000 masl. This analysis shows that Canadian Taiga and Alaskan Boreal Interior are the predominant sources of boundary layer CO emissions reflecting fires in these regions at that time. It should be noted that large enhancement values for $CO_2$,
$CH_4$ and CO were observed with the Alaskan Boreal Interior, which were the result of samples taken in the early morning (10:00 local time) before the boundary layer had fully developed (typically around 11:00-12:00 local time). This trapping of night-time emissions results in significant surface enhancements that quickly taper off with altitude. These measurements were typically taken during the first profile out of Fairbanks where the majority of the Arctic-CAP flights originated.

3.2.    Model Data Comparisons

Aircraft profiles that measure the gradient from the boundary layer into the free troposphere are particularly useful for evaluating atmospheric models and for separating errors and uncertainties related to atmospheric vertical transport and surface flux model simulations. This is demonstrated by

comparing surface flux models for $CO_2$, $CH_4$, and CO using a single GCM to evaluate the land surface flux model.

### 3.2.1.     Point by Point Comparison

In the GEOS model run used for these comparisons, an effort was made to match the global atmospheric burdens of $CO_2$, $CH_4$ and CO; however, given the uncertainties in the sources and sinks of these trace gases and in the representation of long-range and local atmospheric transport, it is not uncommon to have mean offsets between the observed and the modeled mole fractions. To evaluate surface fluxes in the ABoVE domain, it is important to consider both the impact of regional-scale fluxes and long-range transport processes that control the mole fractions of $CO_2$, $CH_4$ and CO throughout the ABoVE domain. A time series comparison of the modeled and the observed $CO_2$, $CH_4$ and CO mole fractions (Fig. 6) suggests that gross features of the seasonal cycles are matched, although some significant differences require detailed analysis by considering different elements of each vertical profile.

### 3.2.2.     Free Troposphere Comparisons

As demonstrated from the analysis of the boundary layer enhancements (Fig. 6) observed during Arctic-CAP, it is useful to subtract the average free tropospheric mole fraction from each profile to better understand the local influences within a particular profile. Differences in the mean free tropospheric values, however, can be a valuable indicator of how large-scale biases in the model influence point-to-point comparisons.

In the case of $CO_2$, the mean daily $CO_2$ mole fraction in the observed free troposphere is increasing faster than modeled values over the course of 6 research missions. The largest offset exceeds a mean value of ~2 ppm (observed – modeled) during the September campaign (Fig. 7). Based on the available model runs, it is difficult to diagnose what causes this offset, although a few hypotheses can be put forward. Given the decreasing latitudinal gradient for $CO_2$ in the free troposphere at this time of year, the offset could be explained by sluggish meridional transport in the model. Alternatively, exaggerated biological uptake in the model in regions outside the study area could be pulling down the $CO_2$ in modeled free troposphere more rapidly than the drawdown observed over the ABoVE domain.

Likewise, measured $CH_4$ increases faster than modeled $CH_4$ over the course of the campaign. Given the decreasing meridional gradient for $CH_4$ that exists during the summer months, sluggish transport could explain the difference between model and observations. Alternatively, modeled June-July-August emissions of $CH_4$ in areas contained by the ABoVE domain could be underestimated, leading to slower increase in modeled free tropospheric $CH_4$.

Finally, the difference between modeled and observed mole fractions of CO in the free troposphere is mainly driven by inaccuracies in the modeled CO from fire plumes both within and outside the ABoVE domain. Figures 4, 6 and 7 show observations of large CO enhancements above 4000 masl during the July, August and October/November campaigns. Local fires were likely responsible for the large excursions in the free tropospheric CO between different profiles. Accurately simulating the injection height of fire plumes is challenging (Freitas et al., 2007; Strode et al., 2018). The GEOS model

distributes biomass burning emissions throughout the planetary boundary layer (PBL) to represent injection above the surface layer, but this method can result in underestimated local emissions for fire plumes detraining in the free troposphere. In regions remote to the ABoVE domain, emissions can be
mixed and lofted by large-scale weather systems, which may explain why the model performs better in simulating long-range CO plume transport than it does in capturing the CO enhancements from local fires. The observation-model mismatch is likely compounded by the inability of the model to accurately simulate the subgrid-scale vertical mixing necessary for capturing vertical profiles for local sources.

### 3.2.3.  Boundary Layer Comparisons

Accurately modeling boundary layer mole fractions of $CO_2$, $CH_4$ and CO depends on an correct representation of two key factors. First, there is a need to accurately model the local surface-atmosphere flux and second there is a need to correctly model the physical evolution of the PBL, as well as horizontal transport and vertical mixing out of the PBL into the free troposphere. GCMs have limited horizontal and vertical resolution and require parameterizations to predict both the rate of change and
the absolute value of the PBL height over the course of the day. Errors in PBL mixing directly impact the tracer mole fraction estimate. Overestimation of the PBL height causes an artificial dilution of the impact of surface flux. Conversely, underestimation of the PBL height results in amplification of the impact of a surface flux on the simulated PBL mole fraction. Additionally, GCMs typically simulate large-scale horizontal gradients more accurately than PBL height unless there are large topographic
changes that occur on horizontal scales less than the model resolution (for GEOS, 0.5 degree). This is because such large-scale patterns are generally well-constrained by the millions of in situ and satellite observations incorporated into meteorological analyses while PBL mixing is represented by highly simplified parameterizations
The three carbon species that we investigate in this study provide different diagnostic information about
the model transport and flux specifications. In the case of a gas like CO that often comes from a specific point source in the Arctic, accurate placement of the emissions, both in the horizontal and the vertical, and the modeled wind direction are critical factors. The ABoVE domain is made up of large expanses of forest and tundra in which $CO_2$ fluxes are more uniformly distributed, making the transport accuracy of individual plumes a less critical factor for simulating $CO_2$. Accurately estimating $CH_4$ mole fractions
may be more sensitive to horizontal transport in the PBL if $CH_4$ emissions are dominated by specific features such as lakes or wetlands, or anthropogenic point sources from oil and gas production such as those observed on the North Slope (Floerchinger et al., 2019). However, we observed consistent PBL $CH_4$ enhancements throughout each campaign (Fig. 5), suggesting a spatial homogeneity in $CH_4$ emissions rather than emissions from specific point sources.

3.3.  Altitude-integrated Enhancements (AIEs)

While individual mole fraction measurements are challenging to reproduce given errors in both modeled surface fluxes and transport, the vertical profile provides a unique opportunity for removing significant uncertainties in transport in order to better assess the surface flux model of a specific long-lived tracer. Assuming that horizontal transport is a relatively small source of bias and the upper part of the free
troposphere (>3000 masl) is largely unaffected by local processes, it is possible to use the information

in the vertical profile to reduce the effects of vertical transport. This can be estimated by vertically integrating the net change in the PBL due to a surface flux from the surface to a specific altitude that is well above the boundary layer. For this study, almost all the enhancements for $CO_2$ and $CH_4$ were observed below 3000 masl.

By subtracting the average free tropospheric (FT) values in both the model and the measurements and averaging the resulting enhancements or depletions for each profile mapped on equal altitude bins from surface to 3000 masl (Eq. 1), we quantify a total enhancement (AIE) resulting from the surface flux (Fig. 8). The resulting measured and modeled AIE show good correlations for $CO_2$ and $CH_4$ but the CO correlations are not as promising.

The average measured enhancement in $CO_2$ and $CH_4$ below 3000 masl is correlated with the forward model such that more than 50% and 36%, respectively, of the observed variability is captured by the model (Fig. 8). The average CO enhancements in the lower 3000 masl is captured by the model with lesser accuracy – in fact, the model only captures 26% of the observed variability along with a significant bias throughout the growing season.

### 3.3.1.       $CO_2$ AIE

To understand the true value of the aircraft profile in evaluating the ability of the surface flux model to reproduce observed fluxes over large regional expanses, it is useful to rigorously compare the differences between modeled and observed near-surface enhancements. The enhancements of $CO_2$ below 3000 masl shown in Fig. 8 for both data and the GEOS model are well correlated. As expected,

during April/May we see very little change in the AIEs below 3000 masl, while June and July and August show significant drawdown, followed by enhancements in September and October/November (Fig. 6 and 8). The modeled AIEs in the lower 3000 masl reproduce the observations suggesting that the surface flux of $CO_2$ throughout most of the ABoVE domain is accurately modeled by GEOS. Despite the overall agreement indicated by aggregated statistics, a closer look shows significant

differences in observed and modeled $CO_2$ enhancements for many individual flight days (Fig. 9). Inspection of individual profiles (Fig. 10) reveal that in some cases the model is not capturing near-ground stratification observed in the river valleys of the interior parts of the ABoVE domain. This is not surprising given that the observations have a much higher vertical resolution than the model's vertical resolution, which is ~100m in the PBL. Consequently, the observed mole fraction values are much

higher than the model estimates because the model is not able to capture the stratification. However, the overall modeled vertical gradients in $CO_2$ match the observations suggesting that the large-scale vertical transport of emissions is accurately simulated above ~1000 masl. As an example, the set of profiles from July 10 (Fig. 10) demonstrates that, although infrequent, high PBL heights and emissions from fires (as indicated by large (>400 ppb) enhancements in CO) add some uncertainty to the AIE values.

Both of these factors impact the mean free tropospheric correction and altitude of integration that we have chosen to accurately capture the total $CO_2$ enhancement from the surface fluxes.

### 3.3.2.    CH₄ AIE

Although the correlation between the observed and modeled AIEs of $CH_4$ is significant, they are not as good as they are for $CO_2$. In particular, we see some clear biases in the seasonality where the
enhancements in the early part of the season are underestimated by the model while the enhancements in the later part of the season are overestimated. This is demonstrated both by the comparisons of AIEs (Fig. 8) and of mole fraction enhancements below 3000 masl (Fig. 9) where the mean difference (observed – modeled) switches from positive to negative over the course of the study period. The Arctic-CAP profile observations provide a critical point of comparison to which future surface flux
models of $CH_4$ can be compared, helping to identify areas where process improvements are needed.

### 3.3.3.    CO AIE

The comparison of observed and modeled AIEs of CO is less useful because some of the critical assumptions made for this comparison are designed to shed light on surface processes affecting $CO_2$ and $CH_4$. The biggest limitation in the CO simulation for interpreting vertical profile observations appears to
be in the accuracy of the vertical distribution of CO emissions. While the model shows an increase in mole fractions during the July and October/November campaigns, the extreme mole fractions in the observations are twice that of the model (Fig. 6). A good example of how the model and the observed mole fractions are different can be seen on July 10, 2017 (Fig. 10) during a flight up the Mackenzie River in the Northwest Territories of Canada. Here, large enhancements of CO (>400 ppb) are observed
at altitudes between 3000 and 5000 masl while $CH_4$ and $CO_2$ boundary layer enhancements are observed below 3000 masl in most of the profiles measured that day. The ~100 ppb CO/ppm $CO_2$ ratio and the large CO enhancement not only support the idea that a fire is the source but that the fire is nearby (<100 km). Both the magnitude and altitude of the CO enhancement point to a few critical limitations in the model that was less important for $CO_2$ and $CH_4$. First, most GCMs, including GEOS,
do not take into account the massive heat source that fires provide to correctly model the injection of fire emissions above the boundary layer. Second, the fire radiative power observations used to estimate emissions can be obscured by thick clouds or aerosols resulting in the emissions estimates missing some fire hotspots. Third, the heterogenous nature of fires as a surface source of CO means that any inaccuracies in horizontal transport or location of the fire will play a large role in the ability of the
model to accurately reproduce the observations. Fourth, the lack of diurnal cycle in biomass burning emissions from the emission database (QFED; Table 1) may result in 'temporal aggregation errors', whereby the model simulations may miss the high emission values that coincide with the daytime aircraft observations.

### 3.3.4.    Model-data mismatch over ecoregions

The bulk quantity AIE can be used to evaluate surface flux models with aircraft profiles at the regional-scale (Fig. 11). For most regions and times of year, the difference in $CO_2$ AIEs is not statistically significant; however, there are certain regions such as the Northern Tundra of Alaska, where the modeled $CO_2$ AIEs are significantly different and amplify a pattern that is observed over other regions. In early spring, the model slightly overestimates observed boundary layer enhancements but a month

later the model underestimates drawdown. Figures 6 and 11 suggest that the peak in early-summer model drawdown in $CO_2$ is preceding the observed $CO_2$ drawdown. The difference between observed and modeled enhancements change sign again during the July flight in Northern Tundra Alaska with an underestimation of the drawdown. Similar patterns can be observed in the Canadian Boreal Cordillera, suggesting that the timing of the summertime drawdown is too early in the model in this region. Over the same period, however, comparisons over the Western Alaska Tundra depict opposite patterns (although far more subtle). While the offsets in the fall months are smaller, there is the suggestion that the enhancements in the Southern Arctic and Canadian Taiga ecoregions are both underestimated in the model. For $CH_4$, the seasonal bias (underestimation in the spring and overestimation between July-September) in the AIEs between observations and models stands out as the most significant feature. The notable exceptions are again the Northern Tundra of Alaska and Canadian Boreal Cordillera, where $CH_4$ AIEs in July and at the end of October are significantly underestimated. For reasons explained earlier, the CO comparison is less informative. However, if one were to analyze data from the month of September, which had no significant influence from fires in the free troposphere, it would suggest that the model continues to underestimate the impact of CO emissions across all regions.

### 3.3.5. Separating local, region and global vertical gradients

By extracting enhancements below 3000 masl from the observations and the model we have largely separated two major sources of biases and uncertainty in a model-data comparison – vertical transport and offsets in background mole fraction. However, it should be acknowledged that gradients between the boundary layer and free troposphere are not controlled exclusively by local fluxes and that in the Arctic, in particular, vertical gradients can be controlled by non-local influences. To explore the impact of long-range transport Parazoo et al. (2016) preformed three simulations to better understand the drivers of the vertical gradient over Alaska and found that 48% of the amplitude (April/May-July/August) in the seasonal vertical gradient was driven by local fluxes from Alaska while the rest was driven by fluxes from the rest of the Arctic (11%) and low latitude (<60N, 41%). For $CO_2$, the impact of long-range transport to the vertical gradient is complicated by the difference in timing of the initial drawdown in the spring and the uptick in the fall at low latitudes verses that of high latitudes. The earlier drawdown of $CO_2$ at low latitudes and the transport of that air via the free troposphere to Arctic significantly reduces the negative vertical gradient in the Arctic. At the same time, the early uptick of $CO_2$ mole fraction in the Arctic relative to the low latitudes enhances the positive vertical gradient in the early fall (Parazoo et al., 2016).
To account for the background vertical gradient in $CH_4$ entering the contiguous US, Baier et al. (2020) and Lan et al. (2019) subtracted 12-15 ppt from the vertical gradient to account for a preexisting gradient in $CH_4$ coming onto the continent. Analysis of the background gradient suggests that this preexisting vertical gradient is a combination of upstream emissions and wind shear which separates the origin of the boundary layer air from that of the free troposphere. Large meridional gradients in $CH_4$, such as those observed in the mid latitudes, will drive depletion of the free troposphere relative to that of the boundary layer over the Arctic. Similarly, CO vertical gradients will also be affected by non-local fluxes and wind shear between the boundary layer and the free troposphere. In the case of CO and $CH_4$ there is also likely to be a vertical gradient that is influenced by the oxidation of these molecules.

505 However, given the relatively long residence time of these molecules and the low sampling altitude in the free troposphere (between 3000 and 5000 masl) of this experiment, this effect is small. From this perspective, the preexisting vertical gradient outside the domain of interest illustrates the importance of the model accuracy in non-local fluxes and the importance of long-range transport in the analysis. One approach ensuring a better boundary conditions is to use a global inversion (e.g. 510 CarbonTracker (Peters et al., 2007)) to initialize the local region where the prognostic flux model is then run to simulate local fields as is done to initialize regional Legrangian inversion models (e.g. Hu et al., 2019).

### 3.3.6.    AIEs as a tool for benchmarking fluxes

This comparison of AIEs from Arctic-CAP and GEOS demonstrates one of the many values of the 515 aircraft profiles as metric for evaluating model performance. In a similar vein, Stephens et a.l (2007) used the vertical gradient to evaluate the model performance which pointed significant errors both from the surface flux models and the vertical transport in the Transcom 3 inversions (Gurney et al., 2002; 2004). The AIE approach has also been used extensively in the Amazon and Arctic as means of optimizing fluxes in an inversion framework. Zhou et al. (2002), Miller et al. (2007) and Gatti et al. 520 (2010; 2014) have all used some form of AIE from aircraft profiles to estimate surface fluxes of $CO_2$ and $CH_4$ in the Amazon basin. Similarly, Zhang et al. (2014), Hartery et al. (2018) and Commane et al. (2017) use the AIE to produce a set of optimized fluxes $CH_4$ and $CO_2$ in the Alaska region. This approach to quantifying regional fluxes has significant advantage over other approaches because it less dependent on an accurate simulation of vertical transport and boundary layer height as point out in 525 section 3.2.3. However, even in this instance there is a need to calculate the average influence of the boundary layer enhancements and this can change dramatically depending on the accuracy of the modelled boundary layer height relative to the integration height of the AIE. In the comparison between observed and modelled AIE presented in this study the focus is on benchmarking a given model's ability to reproduce the AIE in different regions and seasons to objectively quantify how this model 530 might do as conditions change as is expected with changing climate. From this perspective the need for an accurate simulation of vertical transport largely disappears because the near-field fluxes are not being computed but just evaluated. The obvious caveat to this approach is that changing climate will bring with it different covariations in temperature, water, radiation and nutrient availability that cannot be reproduced over this time and space domain. While this approach does not replace model bench 535 marking using eddy covariance measurements, it provides an important view of how modelled processes reproduce observations over scales of 1-3 days and 10-100s of kms.

### 4.  Conclusions

The Arctic-CAP campaign was composed of 6 different research missions from April to November 2017. It sampled $CO_2$, $CH_4$ and CO vertical profiles from the surface to 5000 masl across the ABoVE 540 domain in Alaska and Northwestern Canada, covering 6 major Arctic ecoregions. Arctic-CAP airborne surveys included large Tundra and Boreal ecosystems that are the likely sources of large changes in the seasonal cycle of $CO_2$ and have been the subject of great speculation about future emissions of $CH_4$.

Arctic-CAP's $CO_2$, $CH_4$ and CO profiles provide an excellent basis for evaluating the surface flux models used within state-of-the-art atmospheric transport models, and thus, are an important tool for understanding carbon cycle feedbacks. Comparisons of Arctic-CAP $CO_2$, $CH_4$ and CO observations against GEOS model show the following main results. For $CO_2$, the flux model (land and ocean biosphere and fossil fuel) reproduces seasonal and regional depletions and enhancements observed by aircraft profiles after adjusting for small systematic offsets. For $CH_4$, the model simulations agree reasonably well with the observed vertical profiles, but the model underestimates $CH_4$ enhancements in the spring and overestimates it in the fall. Modeled North Slope $CH_4$ is underestimated throughout the measurement period pointing to deficiencies in the wetland flux specifications over this ecoregion. For CO, the comparison between modeled and observed values were confounded by large biomass burning enhancements in the free troposphere that were not captured in the model. Despite these minor shortcomings, the forward model estimates for $CO_2$ and $CH_4$ represent a marked improvement in model-data differences compared to those done previously for CARVE (Chang et al., 2014; Commane et al., 2017). Results and the flux budgets demonstrate that model representation of $CO_2$ and $CH_4$ for northern high-latitude ecosystems have advanced significantly since the state-of-the-science survey by Fisher et al. (2014). Inversions of the Arctic-CAP data using these fluxes as the prior estimate should further refine the flux estimates and the budget for the ABoVE domain. We note that our comparisons used only GEOS forward model values and slightly different model-data mismatches may be obtained by using a different transport model.

This study highlights the value of collocated airborne $CO_2$, $CH_4$ and CO vertical profiles for quantifying model strengths and weaknesses and for benchmarking fluxes over larger spatial and temporal scales than is offered by EC comparisons. Such evaluation information is essential to improve model characterization of both surface-atmosphere fluxes and to improve our confidence in the accuracy of projections of future conditions. We strongly recommend regular, systematic $CO_2$, $CH_4$ and CO vertical profile observations across the Arctic as an important and cost-effective method to monitor the Arctic for abrupt transformations or potential tipping points in the permafrost-carbon system.

## 5.  Data Availability

Arctic-CAP insitu data can be found at https://doi.org/10.3334/ORNLDAAC/1658

## 6.  Sample Availability

N/A

## 7.  Video supplement

N/A

### 8. Supplement link

575 N/A

### 9. Team List

N/A

### 10. Author contributions

580 CS, KM, CM, AC did experimental design. CS, TN, SW, KM carried out experiment. CS, AC, CM, KM, RB, SW, LS, LH helped with manuscript.

### 11. Competing interests

The authors declare that they have no conflict of interest.

### 12. Disclaimer

585 N/A

### 13. Special issue statement

N/A

### 14. Acknowledgements

This research was supported by the NASA Terrestrial Ecology Program award #NNX17AC61A,
590 "Airborne Seasonal Survey of $CO_2$ and $CH_4$ Across ABOVE Domain", as part of the Arctic-Boreal Vulnerability Experiment (ABoVE). A portion of the research presented in this paper was performed at the Jet Propulsion Laboratory, California Institute of Technology, under contract with the National Aeronautics and Space Administration. GEOS model runs and the work of AC was supported by funding from the NASA ROSES-2016 Grant/Cooperative Agreement NNX17AD69A. We thank the
595 Thomas Lauvaux and 2 anonymous reviewers for their hard work in reviewing this manuscript.

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

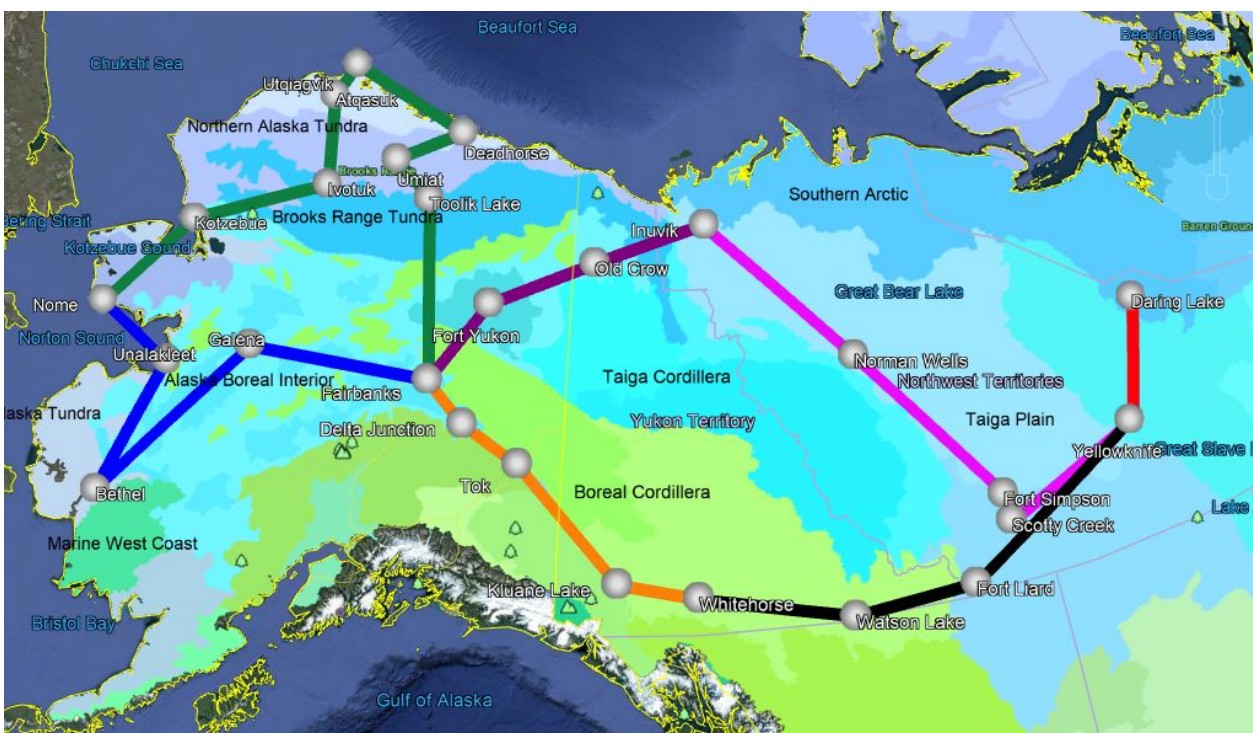

Figure. 1. The Arctic-CAP surveys were designed to sample the Arctic boreal ecosystems of the ABoVE domain. Black text labels represent the six ecoregions covered by this study and white text denote cities and states / provinces. Gray dots depict the locations on which the Arctic-CAP vertical profiles were centered (© Google Earth). Flight track colors represent extent of each (of 7) daily flights (see Figure 2).


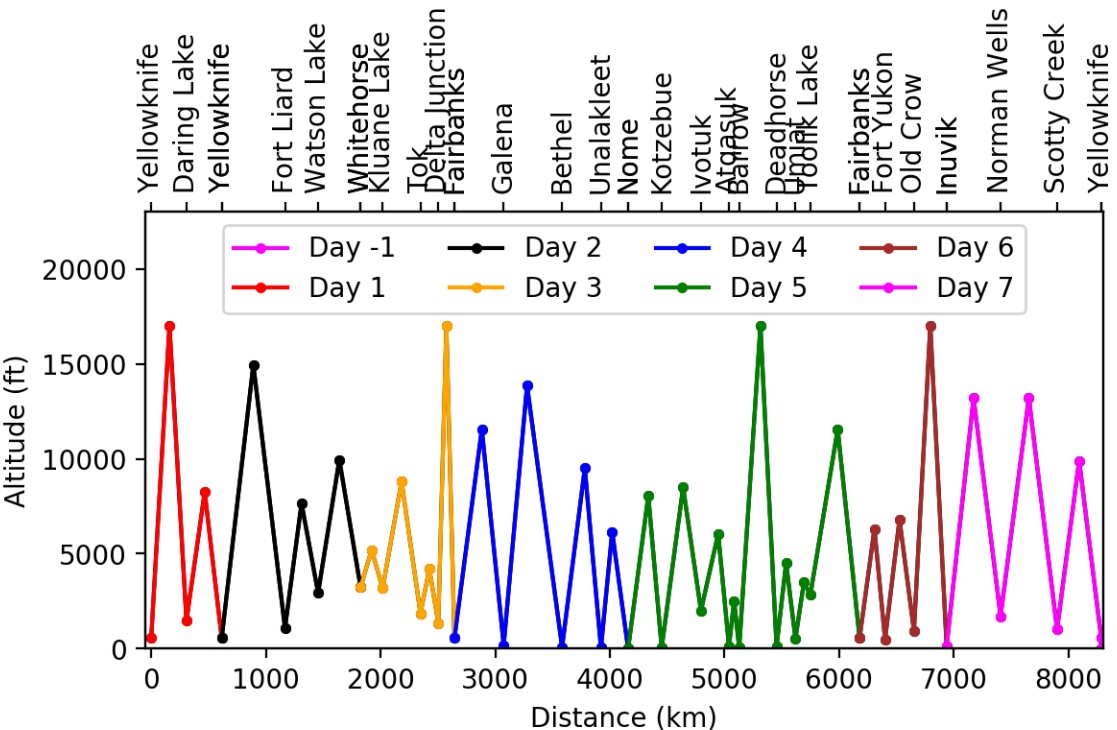

**Figure. 2. Locations and maximum altitudes of the 25 vertical profiles that were acquired during each Arctic-CAP campaign. The colors match the flight lines illustrated in Fig. 1.**

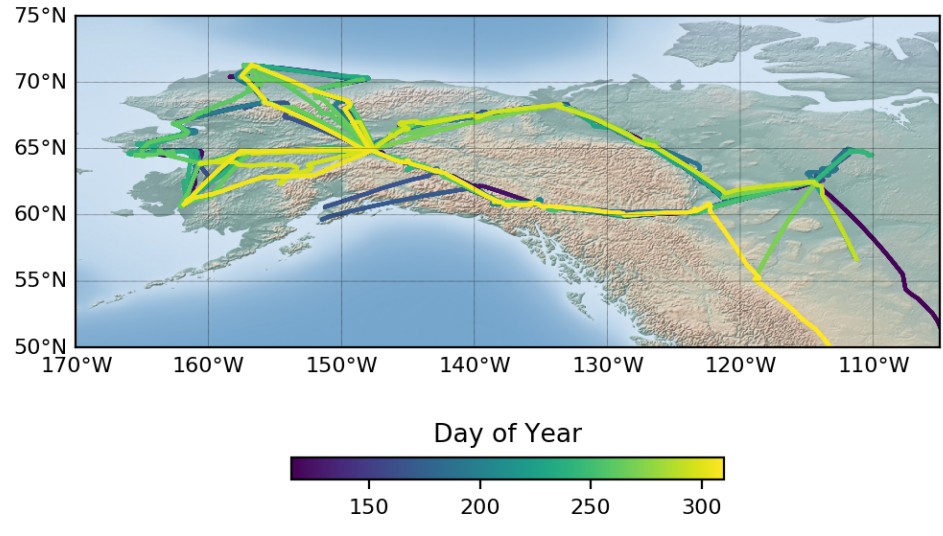


**Figure 3 Arctic-CAP flight paths colored by day of year (DOY). Later paths are plotted on top, masking flights from earlier in the year along the same routes. Profile locations span 50-75 °N and 105-165 °W and sampled environmental conditions from the spring thaw (~DOY 125) through the early cold season (> DOY 300) (© Google Maps).**

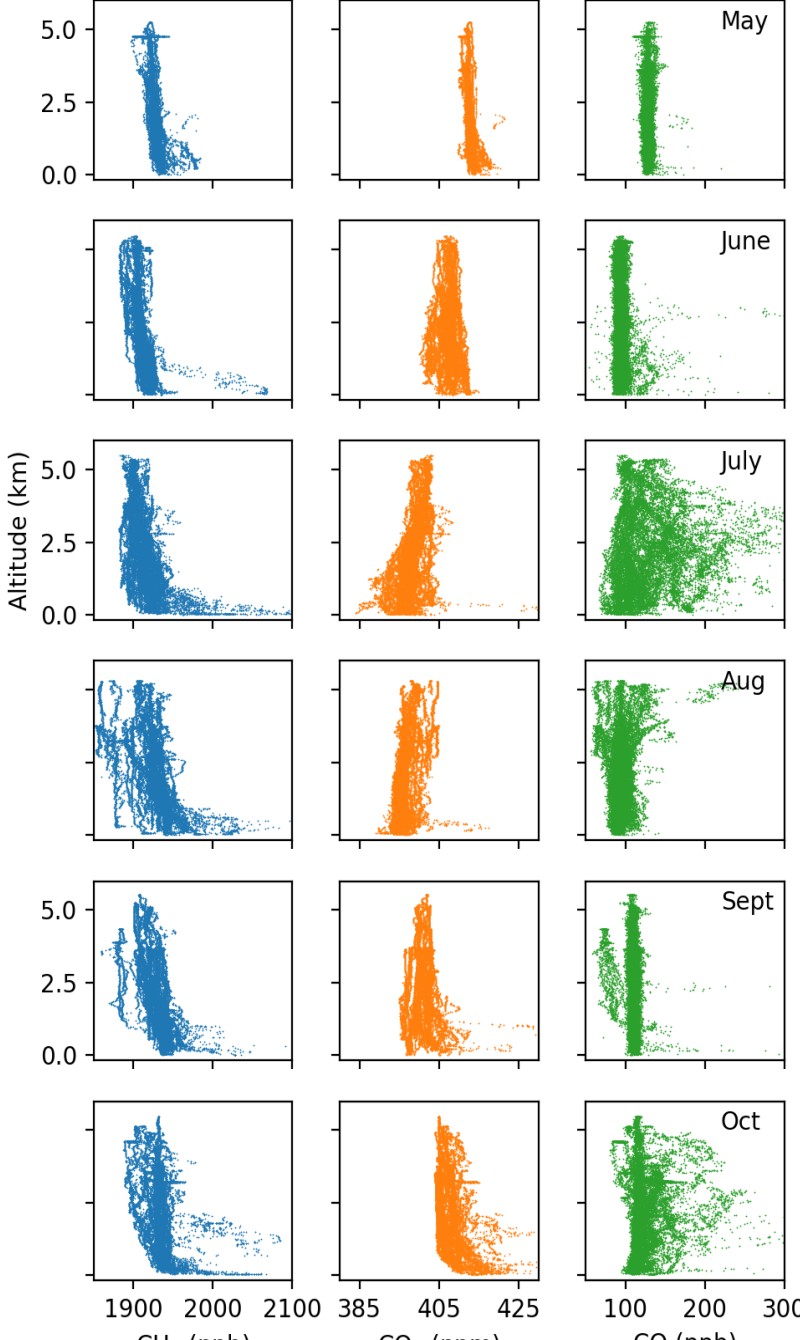


**Figure 4. Composite plots of the CH₄ (left column), CO₂ (center column) and CO (right column) measurements acquired during the Arctic-CAP airborne campaign in 2017. Broad seasonal cycle and near surface enhancement (depletions) can be seen as well as the impact of fires to the free tropospheric CO.**

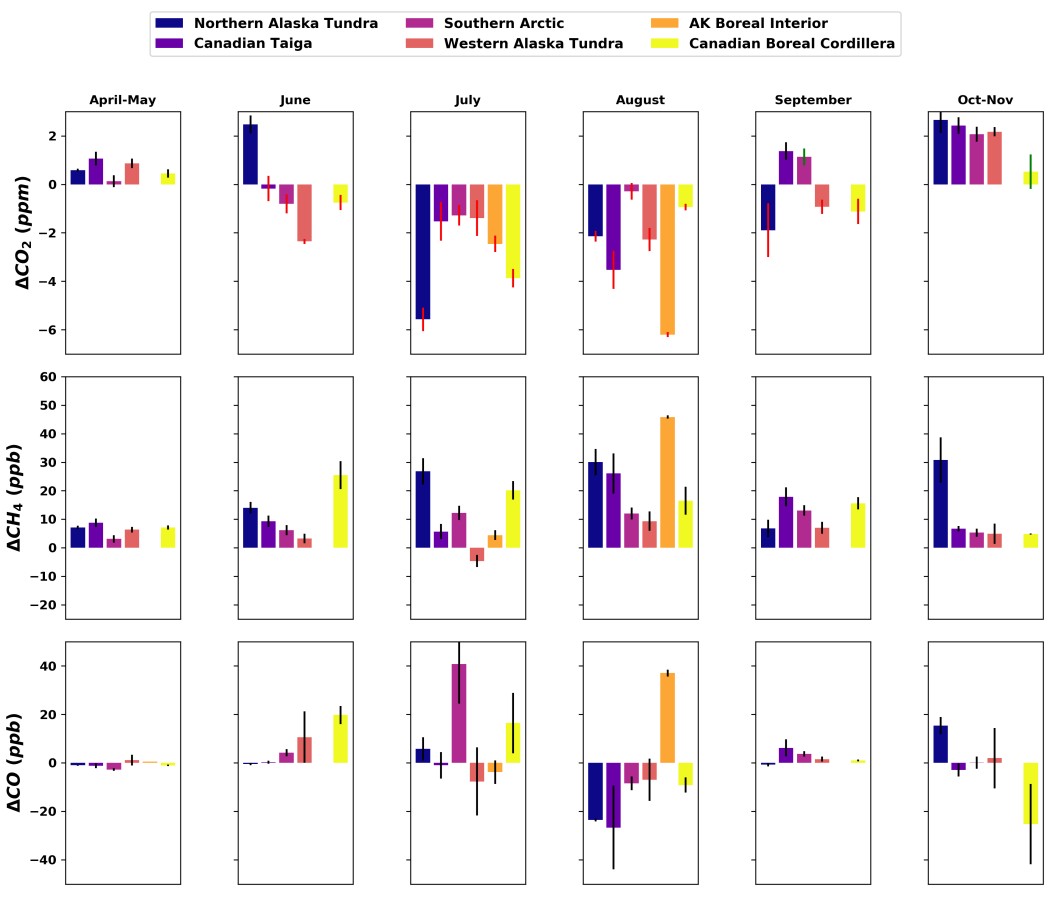


**Figure 5. Average gradient between the mean free daily troposphere (> 3000 masl for CO₂ and CH₄ and 4000 masl for CO) and measurements made below 3000 masl during each campaign. Colors refer to the six ecoregions identified in Fig. 1.**

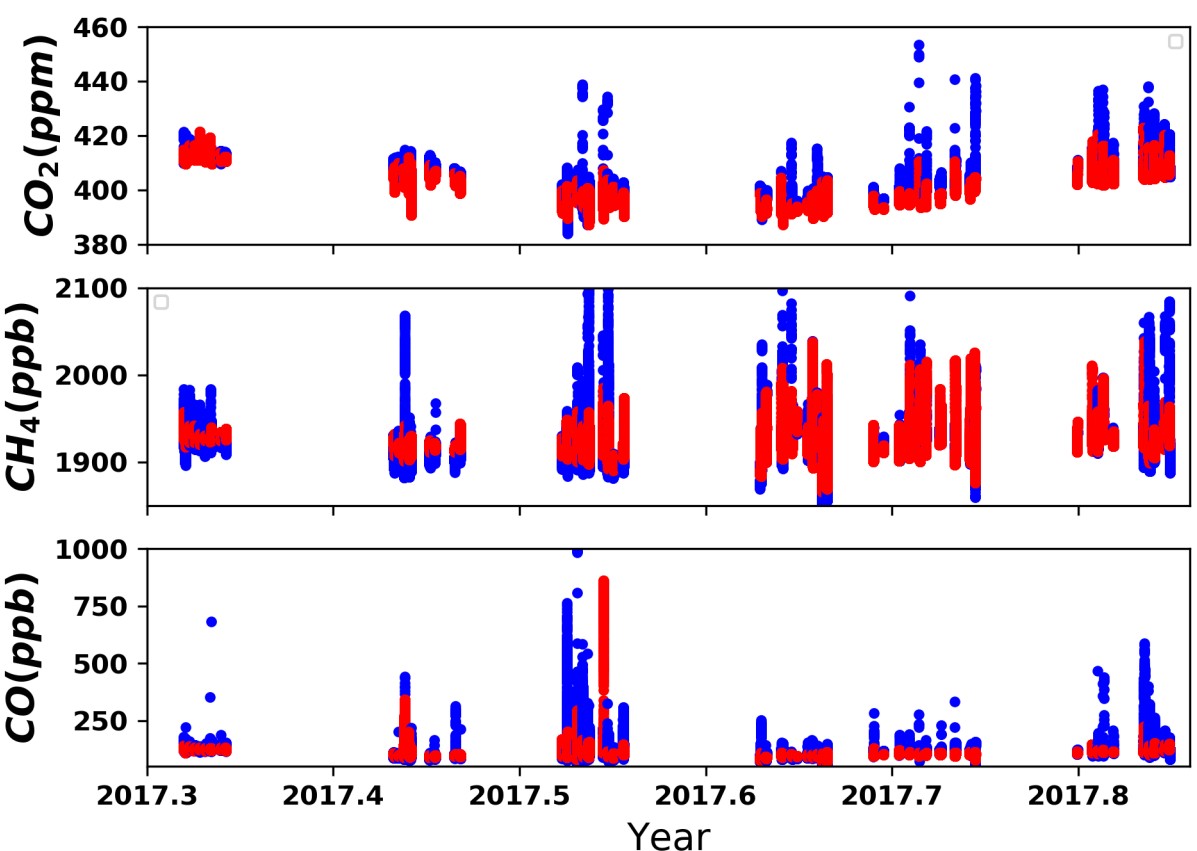

**Figure 6. Comparisons of GEOS simulated atmospheric $CO_2$, $CH_4$ and CO (red points) versus observed $CO_2$, $CH_4$ and CO (blue points) during the Arctic-CAP 2017 campaign show good agreement across campaigns, although the observed data exhibit larger extremes.**

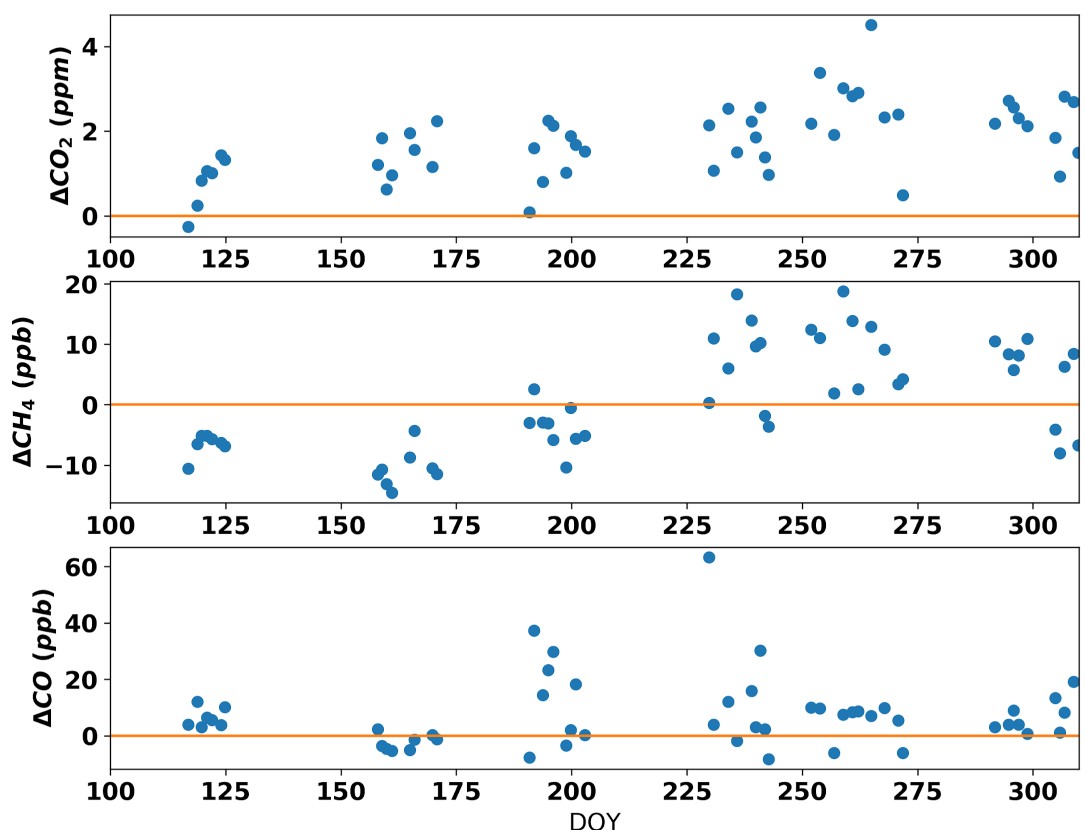

**Figure 7. Difference (observations-model) between mean daily free troposphere (3000-5000 masl for $CO_2$ and $CH_4$ and 4000-5000 masl for CO) for GEOS simulated and Arctic-CAP observed mole fractions. The GEOS simulations systematically underestimate the mean $CO_2$ in all months, while the model overestimates $CH_4$ before DOY 200 and underestimates $CH_4$ after DOY 200. Simulated CO observations generally agree with the atmospheric observations, although there are sporadic underestimates likely associated with incorrectly modeled fire plumes.**


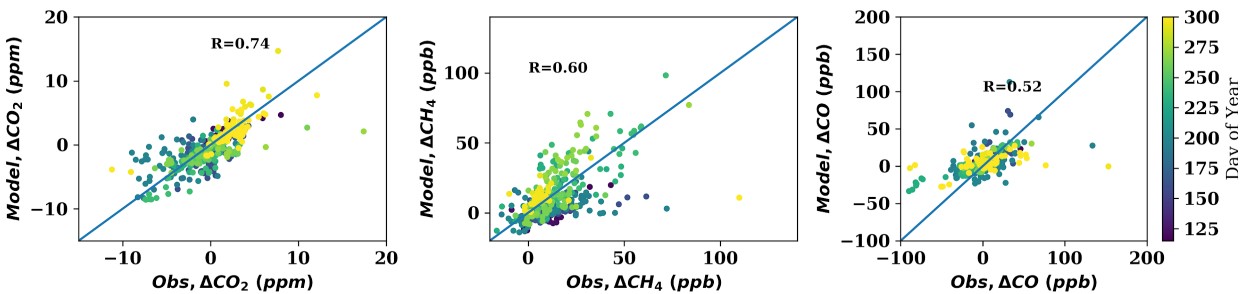

**Figure 8. Modelled versus observed average boundary layer enhancements or depletions in CO₂, CH₄ and CO for individual profiles from 3000 masl down to the surface level.**


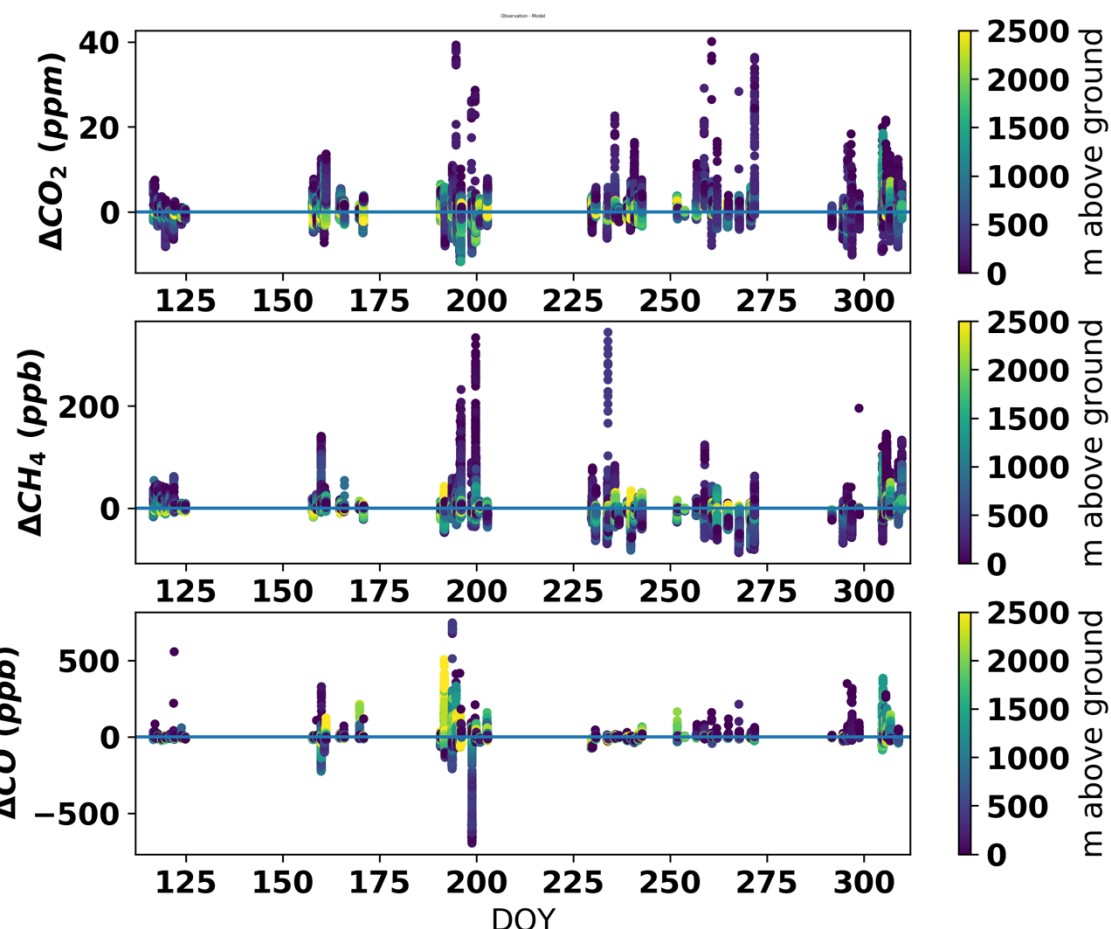

**Figure 9. Observation-model differences in mole fractions below 3000 masl. Corrections have been made for observation-model offsets above 3000 masl (Fig. 7). Colors show the altitude of each deviation. Dark blue indicates differences near the surface while yellow indicates differences near 3000 masl.**


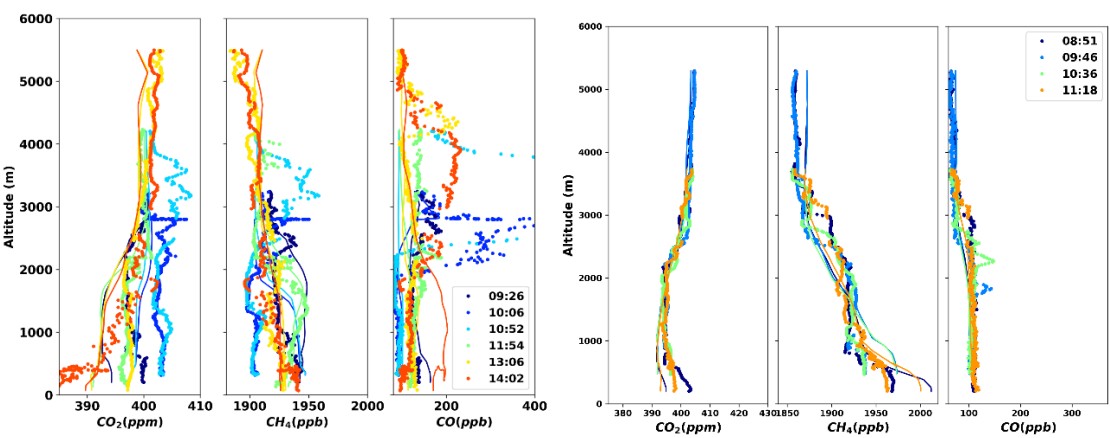

**Figure 10. Observation (dotted lines) and model estimates (thin lines) of profiles on July 10, 2017 (left) and August 30, 2017 (right) from a transect up the Mackenzie River in the Northwest Territory of Canada. Dotted lines show observations and thin lines show model estimates corresponding to specific times during the transect.**


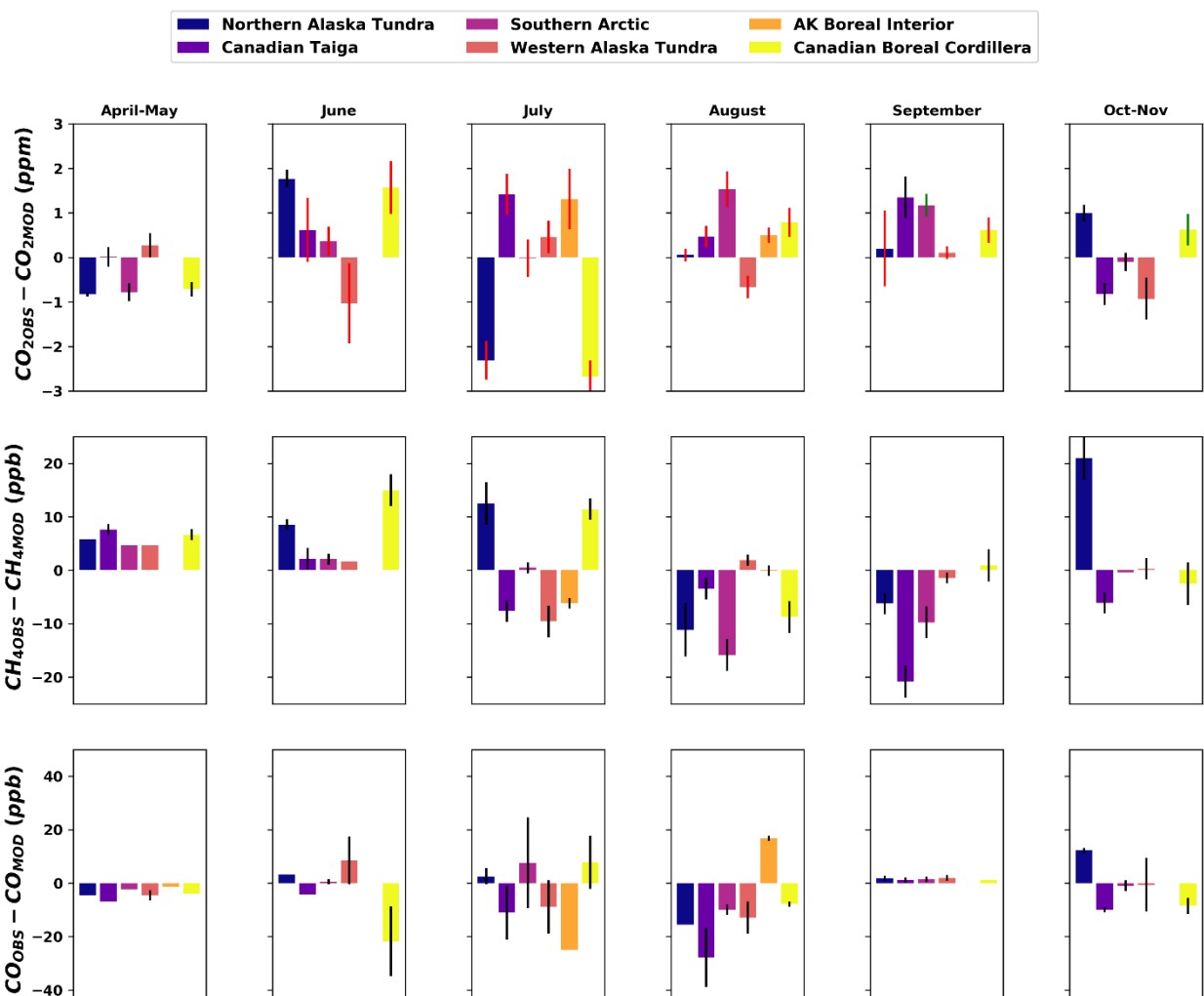

**Figure 11. Average observation–model integrated enhancement differences by ecoregion. Standard deviation of differences for each region are shown with black and red bars. Red (black) bars signify a negative (positive) average enhancement below 3000 meters relative to the daily mean tropospheric value above 3000 masl for $CO_2$ and $CH_4$ and above 4000 masl for CO.**
