# Peer review of "Using atmospheric trace gas vertical profiles to evaluate model fluxes: a case-study of Arctic-CAP observations and GEOS simulations for the ABoVE domain"

_Atmospheric Chemistry and Physics, 2020_

## Referee Comment (RC1) · Anonymous Referee #1 · 28 Sep 2020

The authors discuss recent aircraft observations of trace gases across Alaska and northern Canada and compare these observations against simulations from the GEOS model.

This manuscript is partly a data description paper, partly a model evaluation paper, and partly a paper on trace gas fluxes. I think that a data description paper and model development/evaluation paper would be better-suited for a different journal. At the conclusion of the study, I am not sure what new information we have learned about greenhouse gas sources and sinks from the region. This study seems to reiterate what

we already know – that greenhouse gases exhibit patterns in the atmosphere across North American high latitudes and that atmospheric models can reproduce at least some of these patterns.

Relatedly, I think this manuscript needs a more novel message. I do not think it is sufficient to describe patterns in aircraft observations across northern Canada and Alaska, nor is it sufficient to show that atmospheric models can reproduce at least some of the patterns in those observations. The authors argue that the analysis in this manuscript highlights the potential of the observations to inform surface fluxes, and the authors use a single set of flux estimates to evaluate this question. In fact, a number of studies have used aircraft and tower-based observations across high latitudes to inform surface fluxes (e.g., Pickett Heaps 2011, Chang et al. 2014, Karion et al. 2016, Miller et al. 2016, Zona et al. 2016, Commane et al. 2017, Hartery et al. 2018). Furthermore, existing studies have shown that models can capture spatial and temporal patterns in greenhouse gas fluxes at high latitudes (including models that use GEOS meteorology, e.g., Pickett-Heaps et al. 2011).

It would have been more interesting if the authors would have used these observations to inform surface fluxes instead of commenting on the possibility of doing so. For example, the authors could either estimate fluxes using aircraft observations or evaluate several bottom-up estimates of greenhouse gas fluxes (e.g., from recent bottom-up model inter-comparison projects like the Wetland And Wetland CH4 Inter-Comparison Of Models Project (WETCHIMP), Multi-scale Synthesis and Terrestrial Model Intercomparison Project (MsTMIP), or TRENDY models).

Feedback on specific sections of the manuscript:

Introduction: To date, multiple studies have used atmospheric observations from Alaska and/or Canada to evaluate greenhouse gas fluxes across the region (e.g., Pickett Heaps et al. 2011, Chang et al. 2014, Karion et al. 2016, Sweeney et al. 2016, Zona et al. 2016, Miller et al. 2016, Hartery et al. 2018, Commane et al. 2017, Thompson et al. 2017, Floerchinger et al. 2019). At present, the introduction does not cite any of these studies. I think it's important to cite at least a few of these references, explain what scientific questions these studies were able to answer, and point out which questions remain largely unanswered. The authors point out that top-down studies can be impacted by atmospheric transport errors, but the authors cite studies focused on other regions of the globe, not one of the many studies focused on Alaska and northern Canada.

Lines 195 - 240: The authors compare the magnitude of the bottom-up fluxes used in the GEOS model against a handful of recent studies. There are far more studies that have analyzed carbon dioxide and methane fluxes across the study region than described in the comparison here (see references in previous comments). This material also feels like it would be better suited for a results and/or discussion section; one could evaluate the magnitude of the fluxes using ABoVE aircraft observations and bring in existing studies to provide context on the results of this evaluation.

Section 2: At present, the methods section describes the aircraft flights and the GEOS model simulations, but the methods section does not describe any kind of analysis. I.e., based on the methods section, it is not clear what the authors intend to do with these aircraft observations or model simulations. I think this feeds into my previous comment that the manuscript needs a stronger purpose that drives toward a novel conclusion about greenhouse gas fluxes. I think that including analysis methods in this section would give the reader a stronger preview of where the manuscript is headed and what the authors intend to do with these data and model simulations.

Section 3.1.1: This section provides a broad description of the aircraft flights. I would start the results section by discussing a scientific result. The material in this section almost feels like it would be better suited for a methods section describing the frequency and locations of aircraft flights.

Line 376: Again, I think a number of studies focused on high latitudes have already

shown the utility of using aircraft observations to evaluate surface fluxes and surface flux models. I would try to ask more novel and focused questions either about the information content of existing atmospheric observations or about surface fluxes.

Minor comments: Line 169: Make sure to put the "2" in "CO2" in subscript.

Figure 1: Why are the different flight tracks shown in different colors? What do those colors mean? This information is explained in the Figure 2 caption, but it would be useful to explain these features within the caption for figure 1 (or within a legend).

Figures 4 and 5: What is the takeaway message of this figure? I would recommend adding this information to the caption. Each of these figures contains many panels, and I was not sure what message I should take away as a reader.

Make sure to put the "2" and "4" in "CO2" and "CH4" in subscript throughout the references section.

References:

Chang, R. Y. W., Miller, C. E., Dinardo, S. J., Karion, A., Sweeney, C., Daube, B. C., Henderson, J. M., Mountain, M. E., Eluszkiewicz, J., Miller, J. B., Bruhwiler, L. M. P., and Wofsy, S. C.: Methane emissions from Alaska in 2012 from CARVE airborne observations, Proc. Natl. Acad. Sci. U. S. A., 111, 16694-16699, 10.1073/pnas.1412953111, 2014.

Commane, R., Lindaas, J., Benmergui, J., Luus, K. A., Chang, R. Y. W., Daube, B. C., Euskirchen, E. S., Henderson, J. M., Karion, A., Miller, J. B., Miller, S. M., Parazoo, N. C., Randerson, J. T., Sweeney, C., Tans, P., Thoning, K., Veraverbeke, S., Miller, C. E., and Wofsy, S. C.: Carbon dioxide sources from Alaska driven by increasing early winter respiration from Arctic tundra, Proc. Natl. Acad. Sci. U. S. A., 114, 5361-5366, 10.1073/pnas.1618567114, 2017.

Floerchinger, C., McKain, K., Bonin, T., Peischl, J., Biraud, S. C., Miller, C., Ryerson, T. B., Wofsy, S. C., and Sweeney, C.: Methane emissions from oil and

gas production on the North Slope of Alaska, Atmospheric Environment, 218, 10.1016/j.atmosenv.2019.116985, 2019.

Hartery, S., Commane, R., Lindaas, J., Sweeney, C., Henderson, J., Mountain, M., Steiner, N., McDonald, K., Dinardo, S. J., Miller, C. E., Wofsy, S. C., and Chang, R. Y.-W.: Estimating regional-scale methane flux and budgets using CARVE aircraft measurements over Alaska, Atmos. Chem. Phys., 18, 185–202, https://doi.org/10.5194/acp-18-185-2018, 2018.

Karion, A., Sweeney, C., Miller, J. B., Andrews, A. E., Commane, R., Dinardo, S., Henderson, J. M., Lindaas, J., Lin, J. C., Luus, K. A., Newberger, T., Tans, P., Wofsy, S. C., Wolter, S., and Miller, C. E.: Investigating Alaskan methane and carbon dioxide fluxes using measurements from the CARVE tower, Atmos. Chem. Phys., 16, 5383–5398, https://doi.org/10.5194/acp-16-5383-2016, 2016.

Miller, S. M., Miller, C. E., Commane, R., Chang, R. Y. W., Dinardo, S. J., Henderson, J. M., ... & Sweeney, C.: A multiyear estimate of methane fluxes in Alaska from CARVE atmospheric observations. Global biogeochemical cycles, 30(10), 1441-1453, 2016.

Pickett-Heaps, C. A., Jacob, D. J., Wecht, K. J., Kort, E. A., Wofsy, S. C., Diskin, G. S., Worthy, D. E. J., Kaplan, J. O., Bey, I., and Drevet, J.: Magnitude and seasonality of wetland methane emissions from the Hudson Bay Lowlands (Canada), Atmos. Chem. Phys., 11, 3773–3779, https://doi.org/10.5194/acp-11-3773-2011, 2011.

Sweeney, C., Dlugokencky, E., Miller, C. E., Wofsy, S., Karion, A., Dinardo, S., Chang, R. Y. W., Miller, J. B., Bruhwiler, L., Crotwell, A. M., Newberger, T., McKain, K., Stone, R. S., Wolter, S. E., Lang, P. E., and Tans, P.: No significant increase in long-term CH4 emissions on North Slope of Alaska despite significant increase in air temperature, Geophysical Research Letters, 43, 6604-6611, 10.1002/2016gl069292, 2016.

Thompson, R. L., Sasakawa, M., Machida, T., Aalto, T., Worthy, D., Lavric, J. V., Myhre, C. L., and Stohl, A.: Methane fluxes in the high northern latitudes for 2005-2013 esti-

mated using a Bayesian atmospheric inversion, Atmospheric Chemistry and Physics, 17, 3553-3572, 10.5194/acp-17-3553-2017, 2017.

Zona, D., Gioli, B., Commane, R., Lindaas, J., Wofsy, S. C., Miller, C. E., ... & Chang, R. Y. W.: Cold season emissions dominate the Arctic tundra methane budget. Proceedings of the National Academy of Sciences, 113(1), 40-45, 2016.

---

## Author Comment (AC1) · 30 Sep 2020

We thank reviewer 1 for his or her comments and suggestions. We believe an important point of our paper was missed by the reviewer, which we take as an indication that our present manuscript may not be clear on this point. Our understanding of natural emissions of $CO_2$, $CH_4$ and CO is rudimentary at best and is limited by data as well as mechanistic models that will allow us to predict change in the Arctic. A main focus of this paper is to describe an approach for evaluating and demonstrating the accuracy of the $CO_2$ and $CH_4$ surface fluxes in the NASA GEOS model for the Arctic. Our approach

takes advantage of the fact that the atmospheric profiles made during the ArcticCAP mission not only sampled most of the boundary layer but also included both residual layers and backgroud free troposphere. By integrating the enhancement of CO2, CH4 and CO with altitude from surface to 3000 masl from aircraft profiles to calculate a total enhancement in the atmospheric column for direct comparison with a forward model, we significantly reduce biases due to both model transport and background mole fraction that confound inverse estimate of surface fluxes. In fact, the majority of the papers that the reviewer refers to, have identified these types of biases as negatively impacting the accuracy of their estimates. But none of those studies aimed to quantify and diagnose such biases using information from multiple species of carbon, as is done in our paper.

One of the largest biases incurred during inverse estimates are the errors in the mixing layer height. By integrating a total enhancement from the surface to 3000 m (well above the boundary layer) we eliminate the bias that comes from inverting a point concentration measurement [mole fraction x molar density = mole/m^3] by using an integral [mole fraction x molar density x alt = mole/m^2]). This method is particularly useful for the Arctic and the high-latitudes, where forward model prediction is important for developing an understanding of how climate change will drive feedbacks in greenhouse gas fluxes as well as emissions of gases like CO. We have used this approach to show that indeed the selected forward model (GEOS) is quite good at replicating CO2 and CH4 fluxes, a first step toward assessing the capacity of the model to predict future concentrations. A second source of error in inverse estimation, especially for regional inversion models, comes from the boundary conditions that are specified. In this manuscript, we demonstrate the fidelity of a global model that can be used to supply the boundary conditions for three different species. This in itself is unique since such a dedicated study to evaluate the capability of a high-resolution global model over this domain does not currently exist in the literature, and is useful since it demonstrates that the GEOS model can supply the boundary conditions for regional inversion models to quantify the fluxes.

As the reviewer points out, there are many papers that have used aircraft data to infer fluxes using inverse methods or by comparing point measurements with forward model output, but few have leveraged the mole fraction x altitude integral method which avoids major biases in background and transport errors, thereby allowing us to evaluate seasonal and regional biases in fluxes and drive substantive future model improvements. The reviewer rightly points out that the next step is to focus on one tracer (e.g. CO2 or CH4) and compare the altitude integral of that species to the output from multiple bottom-up flux estimates as a way to rank/evaluate their accuracy. This paper focuses on the pros and cons of using this method on three different tracers. We demonstrate that in the Arctic the method might not work for CO due to the nature of the subgrid pyroconvection, and for CH4 we have identified areas where the seasonal and regional emissions need improvement. Because our altitude-integrated enhancement is dominated by local fluxes and is relatively agnostic to modelling errors, it is a quantity that can and should be optimized from a mechanistic perspective - thus improving future model predictability.

---

## Referee Comment (RC2) · Thomas Lauvaux (Referee) · 18 Dec 2020

Review of the paper entitled "Atmospheric carbon cycle dynamics over the AboVE domain: an integrated analysis using aircraft observations (Artic-CAP) and model simulations (GEOS)" by Sweeney et al.

General comment:

This study presents an analysis of aircraft measurements of $CO_2$, CO and $CH_4$ concentrations over Alaska collected in 2017. The comparison to the GEOS modeled

concentrations brings new insights on the current performances of the GEOS global model over high-latitudinal regions. The overall analysis is sound and valid, with interesting findings over sub-regions of Alaska. The data set is highly valuable in a part of the world with a limited number of long-term concentration or ecosystem measurements. However, the analysis falls short with no definitive conclusions related to the surface fluxes or the transport at high latitudes. Some of the wind measurements have not been used, and the link between the vertical profile analysis and the model-data mismatches is unclear. By providing additional analyses of the model-data residuals, this study can improve substantially. Therefore, I recommend the following revisions before publication:

- The analysis of the vertical profiles and the model-data mismatches is mostly disconnected. Hence it remains difficult to conclude if the transport model errors or the surface flux errors are responsible for the observed mismatches. Knowing that GHG vertical profiles can provide the required information, in addition to wind measurements collected during the campaign, the authors have the opportunity to examine more closely the causes of the mismatches. Are the peaks related to large vertical gradient mismatches? Or to long-range transport errors (incorrect winds in the FT)?

- Wind measurement profiles have not been used in this study. Assessing the performances of the GEOS model, especially in the Free Troposphere, would provide additional insights on long-range transport in high-latitudes and the representation of the FT-PBL gradients used here. These measurements are quite valuable and coud be useful to this analysis.

- This study lacks a comparison to the EC flux measurement analyses available in Alaska, or even in situ tower-based concentration measurements. Considering the consistent seasonal variations in the model-data biases across sub-regions of Alaska (for CH4 especially) (Fig. 11), an analysis of EC CH4/CO2 measurements would help understand the findings of this top-down evaluation, or demonstrate the differences in bottom-up and top-down evaluation of surface fluxes. In any case, the comparison to

EC flux results or tower-based concentration measurements would help with the lack of definitive conclusions on the causes of the mismatches (long-range transport, local fluxes, vertical mixing,. . .).

Technical comments:

- P2-L59: What type of scaling techniques are you referring to? Add references related to the use of eddy-flux towers and how they have been used. What about biogeochemical model calibration?

- P2-L64: Define "bottom-up" briefly. Does it include eddy-flux tower measurements?

- P2-L68: you're describing a Bayesian inversion here. What about top-down methods without an explicit prior?

- P2-L73: What do you mean by "hybrid" approach? It seems here that you are comparing a forward transport model to aircraft data. Are you referring to the use of multiple species?

- P5-L181: "not bad" - replace with a more objective criterion.

- Figure 3 is not referenced anywhere in the text.

- P7-L216 and Table 1: It seems that yo haven't modified the EDGAR oil and gas emissions according to Floerchinger et al. (2019). How would it affect the relative contribution of industrial and fossil emissions?

- P8-L255: The determination of background values from measurements in the Free Troposphere assumes that the air masses are similar near the surface and higher in the column to extract a local enhancement, or that the model is able to capture the Free Tropospheric winds. Assuming you have wind measurements available from the aircraft profiles as exlpained in the methods section, it would be useful to check if the model is correct (as discussed in 3.1.4 – P10-L308).

- P9-L281: Profiles of multiple species can be used to evaluate vertical mixing but not

really advection or large-scale convection.

- P9-Section 3.1.2: this paragraph should be moved to the method section.

- P9-Section 3.1.3: Most of this paragraph belongs to the method section. The figure (Fig. 6) is hard to read and the conclusions are quite vague and limited to one sentence. It seems that focusing on the model-data mismatches would be more informative. This figure should be improved or moved to SI.

- P9-L305: it also looks like there is a trend in the offset for CO2 and CH4.

- P10-L310: The use of wind measurements might help identify transport model errors here.

- P10-Section 3.1.5: Similar comment than for 3.1.2 and 3.1.3. This section summarizes general knowledge on how to interpret PBL measurements. It should be moved to the method section, or the elements should be discussed later in the paper related specifically to the analysis of your measurements.

- P11-L357-369: this paragraph should be moved to the method section.

- Title of 3.1.6: The title indicates a commentary more than a result. Modify to clarify your findings.

- P11-L368: As you showed earlier, the enhancement is also related to the air mass in the Free Troposphere. It might not be an "enhancement", meaning only due to local surface fluxes. It might be a combination of the differences in the PBL and FT background as well.

- P12-L388: You need to link the model-data mismatches and vertical gradient mismatches. The analysis of Figure 9 needs additional insights and investigations. A figure showing the relationship between incorrect vertical mixing and large model-data mismatches would help identify the role of PBL height mismatches. Can it explain the peaks in CO2 and CH4 in Figure 9?

- Section 3.1.8: This paragraph should be merged with the earlier analysis of Fig 8 and 9. Did EC flux analysis suggest the same (i.e. seasonal offset)?

- Section 3.1.10: This analysis is the the most interesting of the paper. With your data set, it seems possible to separate the impact of the vertical mixing from the fluxes. An analysis of the vertical profiles would allow you to identify vertical mixing issues. An analysis fo the sub-region would also be informative, related to the ecosystems.

- P13-L450: The CH4 model-data mismatches seem consistent across regions (from postive to begative to near-zero). It seems plausible that CH4 EC flux data would give you a similar conclusion, and if they don't, it would also be interesting to point out why, and discuss the value of the aircraft data. In any case, the addition of EC data would be helpful to confirm the results or to show the differences between top-down and EC data analyses, as noted earlier in this study.

- P13-L462: performed

- P13-L460: A succinct description of "three simulations" is missing.

- Section 3.1.11: More a discussion than a result section.

- P14-L487: Lagrangian

---

## Author Response (AR1)

**Response to reviewers**:

*Reviewer #1*:
*The authors discuss recent aircraft observations of trace gases across Alaska and northern Canada and compare these observations against simulations from the GEOS model. This manuscript is partly a data description paper, partly a model evaluation paper, and partly a paper on trace gas fluxes. I think that a data description paper and model development/evaluation paper would be better-suited for a different journal. At the conclusion of the study, I am not sure what new information we have learned about greenhouse gas sources and sinks from the region. This study seems to reiterate what paper we already know – that greenhouse gases exhibit patterns in the atmosphere across North American high latitudes and that atmospheric models can reproduce at least some of these patterns. Relatedly, I think this manuscript needs a more novel message. I do not think it is sufficient to describe patterns in aircraft observations across northern Canada and Alaska, nor is it sufficient to show that atmospheric models can reproduce at least some of the patterns in those observations. The authors argue that the analysis in this manuscript highlights the potential of the observations to inform surface fluxes, and the authors use a single set of flux estimates to evaluate this question. In fact, a number of studies have used aircraft and tower-based observations across high latitudes to inform surface fluxes (e.g., Pickett Heaps 2011, Chang et al. 2014, Karion et al. 2016, Miller et al. 2016, Zona et al. 2016, Commane et al. 2017, Hartery et al. 2018). Furthermore, existing studies have shown that models can capture spatial and temporal patterns in greenhouse gas fluxes at high latitudes (including models that use GEOS meteorology, e.g., Pickett-Heaps et al. 2011). It would have been more interesting if the authors would have used these observations to inform surface fluxes instead of commenting on the possibility of doing so. For example, the authors could either estimate fluxes using aircraft observations or evaluate several bottom-up estimates of greenhouse gas fluxes (e.g., from recent bottom-up model inter-comparison projects like the Wetland And Wetland CH4 Inter-Comparison Of Models Project (WETCHIMP), Multi-scale Synthesis and Terrestrial Model Intercomparison Project (MsTMIP), or TRENDY models).*

We thank reviewer 1 for his or her comments and suggestions. We believe an important point of our paper was missed by the reviewer – we take this as an indication that the original version of our manuscript may not have been clear on this point and we have addressed it in the revised manuscript. Our understanding of natural emissions of $CO_2$, $CH_4$ and CO is rudimentary at best and is limited by data as well as mechanistic models that will allow us to predict change in the Arctic-Boreal region. A primary aim of this paper is to describe an approach for evaluating and demonstrating the accuracy of $CO_2$ and $CH_4$ surface fluxes that are specified in global Earth System models, like the NASA GEOS model, over the Arctic. Our approach takes advantage of the fact that the atmospheric profiles made during the Arctic-CAP mission not only sampled most of the boundary layer but also included both residual layers and background free troposphere. By integrating the enhancement of $CO_2$, $CH_4$ and CO with altitude from surface to 3000 masl from aircraft profiles to calculate a total enhancement in the atmospheric column for direct comparisons with a forward model, we significantly reduce biases due to both incorrect representation of atmospheric transport dynamics and background mole fractions that confound inverse estimate of surface fluxes. In fact, the majority of the papers that the reviewer refers to, have identified these types of biases as negatively impacting the accuracy of their estimates.

None of those studies aimed to identify biases in process-based models – in this paper we demonstrate that identification of biases are indeed possible with aircraft profiles. We argue that this is the novel message of this paper, i.e., we do not always need to implement advanced inverse modeling methods to estimate surface fluxes but as a first and important step, we can use information from aircraft profiles/atmospheric measurements and model simulations to identify and rectify biases in the flux specification and the atmospheric transport dynamics represented in the model.

One of the largest biases incurred during inverse estimates are the errors in the mixing layer height. By integrating a total enhancement from the surface to 3000 m (well above the boundary layer) we eliminate the bias that comes from inverting a point concentration measurement by using an integral. This method is particularly useful for the Arctic and the Northern high latitudes, where forward model prediction is important for developing an understanding of how climate change will drive feedbacks in greenhouse gas fluxes as well as emissions of gases like CO. We have used this approach to show that indeed the selected forward model (GEOS) is quite good at replicating $CO_2$ and $CH_4$ fluxes, a first step towards assessing the capacity of the model to predict future concentrations. A second source of error in inverse estimation, especially for regional inversion models, comes from the boundary conditions that are specified. In this manuscript, we demonstrate the fidelity of a global model that can be used to supply the boundary conditions for three different carbon species. This in itself is unique since such a dedicated study to evaluate the capability of a high-resolution global model over this domain does not currently exist in the literature, and is useful since it demonstrates that the GEOS model can supply the boundary conditions for regional inversion models to quantify the fluxes.

The reviewer rightly points out that the next step will be to focus on one tracer (e.g. $CO_2$ or $CH_4$) and compare the altitude integral of that species to the output from multiple bottom-up flux estimates as a way to rank/evaluate their accuracy. We are exactly doing that, and a couple of different manuscripts are in preparation at this stage. This paper is the precursor to those studies and focus on the pros and cons of using a simple altitude-integrated enhancement method on three different tracers. We demonstrate that in the Arctic there are inherent challenges associated with CO due to the nature of the subgrid pyro-convection. For $CO_2$, our state-of-the-art models may be able to reproduce surface fluxes but may underestimate mole fractions above the boundary layer due to the misspecification of fluxes and/or representation of processes elsewhere, and for $CH_4$ we have identified areas where the seasonal and regional emissions need improvement. Since our altitude-integrated enhancement diagnostic is dominated by local fluxes and is relatively agnostic to transport modelling errors, it is a quantity that can and should be optimized from a mechanistic perspective – thus improving future model predictability.

Finally, keeping in mind the reviewer's comments we have made several additions to the paper to highlight the value of the altitude-integrated enhancement method. We have substantially updated the abstract, introduction and discussion to account for the reviewer's concerns and comments.

Feedback on specific sections of the manuscript:
Introduction:
*To date, multiple studies have used atmospheric observations from Alaska and/or Canada to evaluate greenhouse gas fluxes across the region (e.g., Pickett Heaps et al. 2011, Chang et al. 2014, Karion et al. 2016, Sweeney et al. 2016, Zona et al. 2016, Miller et al. 2016, Hartery et al.*

*2018, Commane et al. 2017, Thompson et al. 2017, Floerchinger et al. 2019). At present, the introduction does not cite any of these studies. I think it's important to cite at least a few of these references, explain what scientific questions these studies were able to answer, and point out which questions remain largely unanswered. The authors point out that top-down studies can be impacted by atmospheric transport errors, but the authors cite studies focused on other regions of the globe, not one of the many studies focused on Alaska and northern Canada.*

We agree with the reviewer that there are many papers that have used direct atmospheric measurements to evaluate and correct for model fluxes using mole fraction measurements. The point of this paper is not to estimate fluxes *per se* but to understand whether state-of-the-art flux models can reproduce observed enhancements (in this case below 3000m) using the altitude-integral of enhancements. Many of the papers referred to above use point measurements to optimize fluxes and thus expose themselves to background and transport errors – we intend to demonstrate a new approach that can help identify and eventually eliminate such biases. We have addressed the reviewer's comments in three ways: 1) more clearly defined the intent of our paper; 2) more clearly defined altitude-integration enhancement method, and 3) we have added some of the references that the reviewer listed where appropriate.
New Text: See new title, abstract and first paragraph of track changes document.

*Lines 195 – 240: The authors compare the magnitude of the bottom-up fluxes used in the GEOS model against a handful of recent studies. There are far more studies that have analyzed carbon dioxide and methane fluxes across the study region than described in the comparison here (see references in previous comments). This material also feels like it would be better suited for a results and/or discussion section; one could evaluate the magnitude of the fluxes using AboVE aircraft observations and bring in existing studies to provide context on the results of this evaluation.*

The reviewer is correct that there is a good argument to have model flux discussion in the results and discussion sections. We agree that is a more standard approach. However, we must first establish why we our showing this comparison – our main goal here is to demonstrate that the GEOS model fluxes are not very different from what past inversion studies have found. Hence our approach here is to simply compare the model results (in terms of fluxes) to a suite of inversion studies. The references cited here are meant to illustrate that. In the discussion section, we make the more important point that the altitude-integrated enhancement method correlates well between observations and model suggesting that these flux models do well against the benchmark dataset that we have created from the Arctic-CAP flights.
**New text:** The average measured enhancement in $CO_2$ and $CH_4$ below 3000 masl is correlated with the forward model such that more than 50% and 36%, respectively, of the observed variability is captured by the model (Fig. 8). The average CO enhancements in the lower 3000 masl is captured by the model with lesser accuracy – in fact, the model only captures 26% of the observed variability along with a significant bias throughout the growing season.

*Section 2: At present, the methods section describes the aircraft flights and the GEOS model simulations, but the methods section does not describe any kind of analysis. I.e., based on the methods section, it is not clear what the authors intend to do with these aircraft observations or model simulations. I think this feeds into my previous comment that the manuscript needs a*

*stronger purpose that drives toward a novel conclusion about greenhouse gas fluxes. I think that including analysis methods in this section would give the reader a stronger preview of where the manuscript is headed and what the authors intend to do with these data and model simulations.*

The suggestion to highlight the method we are using to evaluate the GEOS surface flux by putting it in the methods section is a good one and will help to differentiate our approach from other studies. We have added a new section in the methods describe the altitude-integrated enhancement method.

New Text: AIE calculation

As will be explained in the following results section the surface fluxes of $CO_2$, $CH_4$ and CO in GEOS is compared to aircraft observations by first subtracting the average daily free tropospheric value (>3000 m for $CO_2$ and $CH_4$ and >4000 m for CO, $X_{FT}$) from each measurement below 3000 m and comparing that to the altitude integrated sum

$$\Delta X = \int_{z=ground}^{z=3000} ((X - X_{FT})/n_{BL})\, ndz \qquad\qquad \text{Eq. 1}$$

where $\Delta X$ is altitude-integrated sum of the mole fraction of species X minus $X_{FT}$ divided by the $n_{BL}$ where $n_{BL} = \int_{z=ground}^{z=3000} n\, dz$ and n is the atmospheric number density. It is assumed that the mole fraction of each trace gas species measured at the lowest point in each profile is constant to the ground level. Ground level altitude is taken from USGS (USGS, 2017). Thus, the AIE is equivalent to average enhancement in the boundary layer after accounting for altitude changes in number density.

*Section 3.1.1: This section provides a broad description of the aircraft flights. I would start the results section by discussing a scientific result. The material in this section almost feels like it would be better suited for a methods section describing the frequency and locations of aircraft flights.*

Although the reviewer is correct that the first sentence of this section could go in the methods section it serves to contextualize the rest of the section which talks about the raw profiles that were observed throughout the campaign. For this reason, we believe this subsection should remain in the results section.

*Line 376: Again, I think a number of studies focused on high latitudes have already shown the utility of using aircraft observations to evaluate surface fluxes and surface flux models. I would try to ask more novel and focused questions either about the information content of existing atmospheric observations or about surface fluxes.*

Hopefully the additional clarification that we have made plus a final discussion of "AIE as a tool for benchmarking fluxes" helps to clarify this point.

New Text: This comparison of AIEs from Arctic-CAP and GEOS demonstrates one of the many values of the aircraft profiles as metric for evaluating model performance. In a similar vein, Stephens et al (2007) used the vertical gradient to evaluate the model performance which pointed significant errors both from the surface flux models and the vertical transport in the

Transcom 3 inversions (Gurney et al., 2002; 2004). The AIE approach has also been used extensively in the Amazon and Arctic as means of optimizing fluxes in an inversion framework. Zhou et al. (2002), Miller et al. (2007) and Gatti et al. (2010; 2014) have all used some form of AIE from aircraft profiles to estimate surface fluxes of $CO_2$ and $CH_4$ in the Amazon basin. Similarly, Zhang et al. (2014), Hartery et al. (2018) and Commane et al. (2017) use the AIE to produce a set of optimized fluxes $CH_4$ and $CO_2$ in the Alaska region. This approach to quantifying regional fluxes has significant advantage over other approaches because it less dependent on an accurate simulation of vertical transport and boundary layer height as point out in section 3.3.6. However, even in this instance there is a need to calculate the average influence of the boundary layer enhancements and this can change dramatically depending on the accuracy of the modelled boundary layer height relative to the integration height of the AIE. In the comparison between observed and modelled AIE presented in this study the focus is on benchmarking a given model's ability to reproduce the AIE in different regions and seasons to objectively quantify how this model might do as conditions change as is expected with changing climate. From this perspective the need for an accurate simulation of vertical transport largely disappears because the near-field fluxes are not being computed but just evaluated. The obvious caveat to this approach is that changing climate will bring with it different covariations in temperature, water, radiation and nutrient availability that cannot be reproduced over this time and space domain. While this approach does not replace model bench marking using eddy covariance measurements, it provides an important view of how modelled processes reproduce observations over scales of 1-3 days and 10-100s of kms.

Minor comments:
*Line 169: Make sure to put the "2" in "CO2" in subscript.*
Done.

*Figure 1: Why are the different flight tracks shown in different colors? What do those colors mean? This information is explained in the Figure 2 caption, but it would be useful to explain these features within the caption for figure 1 (or within a legend).*
**New text:** "Flight track colors represent extent of each (of 7) daily flights (see Figure 2)."

*Figures 4 and 5: What is the takeaway message of this figure? I would recommend adding this information to the caption. Each of these figures contains many panels, and I was not sure what message I should take away as a reader.*
New text**: "Broad seasonal cycle and near surface enhancement (depletions) can be seen as well as the impact of fires to the free tropospheric CO."**

Make sure to put the "2" and "4" in "CO2" and "CH4" in subscript throughout the references section.

**Text changes:** Lines 594-830 in track changes

*Reviewer #2*

Review of the paper entitled "Atmospheric carbon cycle dynamics over the AboVE domain: an integrated analysis using aircraft observations (Artic-CAP) and model simulations (GEOS)" by Sweeney et al.

General comment: *This study presents an analysis of aircraft measurements of CO2, CO and CH4 concentrations over Alaska collected in 2017. The comparison to the GEOS modeled C1 concentrations brings new insights on the current performances of the GEOS global model over high-latitudinal regions. The overall analysis is sound and valid, with interesting findings over sub-regions of Alaska. The data set is highly valuable in a part of the world with a limited number of long-term concentration or ecosystem measurements. However, the analysis falls short with no definitive conclusions related to the surface fluxes or the transport at high latitudes. Some of the wind measurements have not been used, and the link between the vertical profile analysis and the model-data mismatches is unclear. By providing additional analyses of the model-data residuals, this study can improve substantially. Therefore, I recommend the following revisions before publication:*
*- The analysis of the vertical profiles and the model-data mismatches is mostly disconnected. Hence it remains difficult to conclude if the transport model errors or the surface flux errors are responsible for the observed mismatches. Knowing that GHG vertical profiles can provide the required information, in addition to wind measurements collected during the campaign, the authors have the opportunity to examine more closely the causes of the mismatches. Are the peaks related to large vertical gradient mismatches? Or to long-range transport errors (incorrect winds in the FT)?*
*- Wind measurement profiles have not been used in this study. Assessing the performances of the GEOS model, especially in the Free Troposphere, would provide additional insights on long-range transport in high-latitudes and the representation of the FT-PBL gradients used here. These measurements are quite valuable and could be useful to this analysis.*
*- This study lacks a comparison to the EC flux measurement analyses available in Alaska, or even in situ tower-based concentration measurements. Considering the consistent seasonal variations in the model-data biases across sub-regions of Alaska (for CH4 especially) (Fig. 11), an analysis of EC CH4/CO2 measurements would help understand the findings of this top-down evaluation, or demonstrate the differences in bottom-up and top-down evaluation of surface fluxes. In any case, the comparison to C2 EC flux results or tower-based concentration measurements would help with the lack of definitive conclusions on the causes of the mismatches (long-range transport, local fluxes, vertical mixing,. . .).*

We thank the reviewer for his or her comments. The reviewer has made some valuable comments that suggest future directions that we could take with this analysis. These comments have helped us to further clarify the main message in our analysis. Our previous version of the manuscript suggested that we were using the Arctic-CAP dataset to better understand transport errors. However, the reviewer is correct in the sense that this would need further analysis and more in-depth analyses of the winds and atmospheric transport dynamics, which is not the aim of this paper. It should also be pointed out that while local wind measurements made from the plane may be helpful in in assessing transport error we would still be left questioning the accuracy of long-range transport dynamics and how well the model simulates them. Similarly, direct

comparison of the EC measurements would require a separate surface sensitivity analysis to match aircraft measurements with EC measurements, which is also not in the scope of this analysis. Instead, we have chosen to keep the focus on the value of the altitude-integration method as a way to benchmark surface fluxes and provide a simple yet effective diagnostic for identifying seasonal and regional biases in the fluxes.

Technical comments:
- *P2-L59: What type of scaling techniques are you referring to? Add references related to the use of eddy-flux towers and how they have been used. What about biogeochemical model calibration?*
=>We thank the reviewer for pointing this out. References (Mekonnen et al., 2016 and Johnston et al., 2014) have been added.

- *P2-L64: Define "bottom-up" briefly. Does it include eddy-flux tower measurements?*
=> Text added: "An alternative to the "bottom-up" evaluation approach which relies on the EC approach"

- *P2-L68: you're describing a Bayesian inversion here. What about top-down methods without an explicit prior?*
=>Reworded to express the idea that we are in general data limited and therefore dependent on some "first guess" about spatial and temporal covariance.
**Text now reads:** An alternative to the "bottom-up" evaluation approach, which relies on the eddy covariance measurements, is the "top-down" approach, which makes use of atmospheric measurements of species like $CO_2$, $CH_4$ and CO and modeled atmospheric transport patterns to infer the surface fluxes needed to reproduce observed atmospheric concentrations (Pickett-Heaps et al., 2011; Miller et al., 2016; Thompson et al., 2017 are examples in the Arctic) over large regional scales. In a data limited region, this inverse approach generally takes a forward-flux model, or a set of observations that are likely correlated with the flux, as a prior or first guess. The inverse approach then estimates the flux by scaling the prior. While the inverse approach results in a flux estimate that meets the constraint of the trace gas measurements and modeled transport, the variability in surface flux from these analyses cannot be directly attributed to mechanisms such as temperature changes, $CO_2$ fertilization, nutrient enrichment and water stress and, therefore do not have any predictive capabilities. Also, inverse methods are influenced by errors in atmospheric transport and assumptions about error covariances, which are difficult to characterize (Gourdji et al., 2012; Lauvaux et al., 2012; Mueller et al., 2018; Chatterjee and Michalak, 2013).

- *P2-L73: What do you mean by "hybrid" approach? It seems here that you are comparing a forward transport model to aircraft data. Are you referring to the use of multiple species?*
=>Have attempted to clarify by stressing that we are not using a simple side by side comparison, i.e., mole fraction measurements. Instead this analysis is using the bulk quantity (Altitude-integrated enhancements), which seeks to eliminate some of the implications of inaccurate transport (vertical transport in particular).
**Text now reads:** In this study, a hybrid approach is taken to evaluate and benchmark the accuracy of current state-of-the-art bottom-up land-surface flux models using a bulk quantity

calculated from atmospheric vertical profiles of trace gas mole fractions. The goal is to present an approach to evaluate land-surface flux models that capture complex carbon cycle dynamics over the northern high-latitudes. NASA's Goddard Earth Observing System (GEOS) general circulation model (GCM) is used with a combination of surface flux components for $CO_2$, $CH_4$ and CO to create 4D atmospheric fields; these fields are subsequently evaluated using the altitude-integrated enhancements (AIE) calculated from profiles collected during the Arctic Carbon Atmospheric Profiles (Arctic-CAP) airborne campaign.

*- P5-L181: "not bad" - replace with a more objective criterion. - Figure 3 is not referenced anywhere in the text. - P7-L216 and Table 1: It seems that you haven't modified the EDGAR oil and gas emissions according to Floerchinger et al. (2019). How would it affect the relative contribution of industrial and fossil emissions?*
=>In Floerchinger et al. (2019), we find very minor fluxes of $CH_4$ relative to natural emissions coming from this region which would not have a major impact on the larger region.

*- P8-L255: The determination of background values from measurements in the Free Troposphere assumes that the air masses are similar near the surface and higher in the column to extract a local enhancement, or that the model is able to capture the Free Tropospheric winds. Assuming you have wind measurements available from the aircraft profiles as explained in the methods section, it would be useful to check if the model is correct (as discussed in 3.1.4 – P10-L308).*
=>The reviewer is right to question the validity of FT as a background . A comparison of the winds aloft between model and data is something that would be worth doing; however, we do not have access to the modelled winds and it is worth noting that this will only give us validation of the winds at the time our measurements but will not enable validation or information about the direction and magnitude of the winds over multiple days. Figure 7 and the discussion in section 3.3.5 address this question.

*- P9-L281: Profiles of multiple species can be used to evaluate vertical mixing but not really advection or large-scale convection.*
=> Good point. Text changed to clarify.
**New Text:** Added "vertical" to "Aircraft profiles that measure the gradient from the boundary layer into the free troposphere are particularly useful for evaluating atmospheric models and for separating errors and uncertainties related to atmospheric **vertical** transport and surface flux model simulations."

*- P9-Section 3.1.2: this paragraph should be moved to the method section.*
=> Good point. We have added a section in the methods to more explicitly define the altitude-integrated enhancement method. Hence, we have left this section in to introduce the rest of the sections.
**New Text:** As will be explained in the following results section the surface fluxes of $CO_2$, $CH_4$ and CO in GEOS is compared to aircraft observations by first subtracting the average daily free tropospheric value (>3000 m for $CO_2$ and $CH_4$ and >4000 m for CO, $X_{FT}$) from each measurement below 3000 m and comparing that to the altitude integrated sum

$$\Delta X = \int_{z=ground}^{z=3000} ((X - X_{FT})/n_{BL})\, n dz \qquad\qquad \text{Eq. 1}$$

where ΔX is altitude-integrated sum of the mole fraction of species X minus $X_{FT}$ divided by the $n_{BL}$ where $n_{BL} = \int_{z=ground}^{z=3000} n\, dz$ and n is the atmospheric number density. It is assumed that the mole fraction of each trace gas species measured at the lowest point in each profile is constant to the ground level. Ground level altitude is taken from USGS (USGS, 2017). Thus, the AIE is equivalent to average enhancement in the boundary layer after accounting for altitude changes in number density.

- *P9-Section 3.1.3: Most of this paragraph belongs to the method section. The figure (Fig. 6) is hard to read and the conclusions are quite vague and limited to one sentence. It seems that focusing on the model-data mismatches would be more informative. This figure should be improved or moved to SI.*
=> The reviewer makes a good point - this was tried in earlier version of the paper, but the point of this text is to help the reader follow along with relevance of each step in the creating the altitude-integrated enhancement approach. Each piece has important information and assumptions that are part of the results and discussion, rather than just being highlighted in the methods section.

- *P9-L305: it also looks like there is a trend in the offset for CO2 and CH4.*
=>Definitely. This is observed and discussed in the previous section in the manuscript.

- *P10-L310: The use of wind measurements might help identify transport model errors here.*
=>Yes, but not in the scope of this analysis (see response above).

- *P10-Section 3.1.5: Similar comment than for 3.1.2 and 3.1.3. This section summarizes general knowledge on how to interpret PBL measurements. It should be moved to the method section, or the elements should be discussed later in the paper related specifically to the analysis of your measurements.*
=>See response to similar comment above.

- *P11-L357-369: this paragraph should be moved to the method section.*
=>A separate section has been added to methods section. See added text above.

- *Title of 3.1.6: The title indicates a commentary more than a result. Modify to clarify your findings.*
=>Now in discussion section. Good point.

- *P11-L368: As you showed earlier, the enhancement is also related to the air mass in the Free Troposphere. It might not be an "enhancement", meaning only due to local surface fluxes. It might be a combination of the differences in the PBL and FT background as well.*
=>Great point. We have addressed this possibility in section 3.3.5.

- *P12-L388: You need to link the model-data mismatches and vertical gradient mismatches. The analysis of Figure 9 needs additional insights and investigations. A figure showing the relationship between incorrect vertical mixing and large model-data mismatches would help identify the role of PBL height mismatches. Can it explain the peaks in CO2 and CH4 in Figure 9?*

=>This demonstrated in Figure 10 where in some cases boundary layer height is captured well while in other instances (July 10, 2017 @14:02 Local), it is not.

*C4 - Section 3.1.8: This paragraph should be merged with the earlier analysis of Fig 8 and 9. Did EC flux analysis suggest the same (i.e. seasonal offset)?*
=>We have added text connecting previous section but wanted to separately address the reason why CH4 enhancements in observations and models were not as well correlated.
**Text Added:** "Although the correlation between the observed and modeled AIEs of $CH_4$ is significant, they are not as good as they are for $CO_2$."
*- Section 3.1.10: This analysis is the the most interesting of the paper. With your data set, it seems possible to separate the impact of the vertical mixing from the fluxes. An analysis of the vertical profiles would allow you to identify vertical mixing issues. An analysis fo the sub-region would also be informative, related to the ecosystems.*
=> Previous text may have been misleading in indicating that was the intent of the paper. We have changed the text to show that we are mainly focused on evaluating the drivers of the difference in the AIE and not individual mole fraction measurements or differences in vertical/horizontal transport.
**Text change:** line 105 used to read "This study uses the bulk quantify from Arctic-CAP aircraft profiles to directly evaluate **both the transport** and the terrestrial surface flux models of $CO_2$, $CH_4$ and CO." and now only reads "This study uses the bulk quantify from Arctic-CAP aircraft profiles to directly evaluate the terrestrial surface flux models of $CO_2$, $CH_4$ and CO."
Line 308 added "vertical" to "for separating errors and uncertainties related to atmospheric vertical transport and surface flux model simulations"

*- P13-L450: The CH4 model-data mismatches seem consistent across regions (from postive to begative to near-zero). It seems plausible that CH4 EC flux data would give you a similar conclusion, and if they don't, it would also be interesting to point out why, and discuss the value of the aircraft data. In any case, the addition of EC data would be helpful to confirm the results or to show the differences between top-down and EC data analyses, as noted earlier in this study.*
=>See initial response to using EC measurements. We think the important point here is that this dataset using the bulk quantity derived from AIE.

*- P13-L462: performed - P13-L460: A succinct description of "three simulations" is missing. - Section 3.1.11: More a discussion than a result section.*
Definitely. We have changed whole section to results and discussion section since there is a lot of discussion throughout what was previously just "results" sections.

- P14-L487: Lagrangian
=>Changed.

---

## Author Response (AR2)

Response to reviewers;

We would like to thank the reviewers for their support of this manuscript and have noted responses to comments and editing suggestions below:

Reviewer 1

Abstract line 2 "Bulk quantities": What does this term refer to? I.e., quantities of what?

**Great catch. This has been clarified.**

Abstract line 3 "demonstrate the utility": Utility for what? I.e., what do these bulk quantities have utility for?

**Sentence reworded**

Abstract line 32 "observed in CO observations": I'd avoid word repetition. How about "evident in CO observations" for "manifest in CO observations"?

**"captured in CO observations"**

Line 27 "Among the most important is the uncertainty we have....": The wording here feels awkward. Here's another option: "Carbon-climate feedbacks are among the most uncertain feedbacks."

Used Carbon-climate feedbacks  (Arora et al., 2020) are among the most uncertain climate feedbacks.

Line 43 "The Arctic, in particular....": I don't think this sentence is grammatically correct. It's not quite a run-on sentence, but the second clause is missing a conjunction and a subject.

**Broke up sentence.**

Reviewer 3:

We note that the review here may have been looking at a different MS because we find non of the references to line numbers to be correct or even quoted material. We have tried our best directly answer the comments despite the fact that references to line numbers  and quotes are confused.

Line 75 The target is 'to evaluate the ability of current land surface flux models', but the conclusions indicate that the goal is not fully achieved – separating contributions of model simulated fluxes from inside and outside the study domain is difficult, as well as separating the impacts of surface fluxes from meteorology and tracer transport. It would be natural to extend evaluation target to a combination of the surface flux models and the transport model.

**We think the original intent of the analysis still stands, in particular, because we have not fully evaluated the transport model.**

Line 224 References may be needed to point to past results, and GCP-CH4 regional estimates are available for comparison.

**Refence to Table containing references has been moved to make this clearer.**

Line 245-250 Justification for the setting of 3000 m as a boundary between PBL and free troposphere was not discussed (any suitable reference?), while the PBL heights diagnosed from potential temperature, humidity, and tracer profiles are often lower than 3000 m. As there is a good volume of evidence on the connection between tracer and potential temperature profiles in the troposphere (Jin et al, 2021), an alternative approach would be to test lower altitude for separator or make it linked to isentropic vertical coordinate.

**A sentence has been added in method section: "As will be explained in the results section, the 3000 m was picked as cutoff for $CO_2$ and $CH_4$ because of the low variability of these tracers above that altitude level where as the cutoff point for CO was chosen to be 4000 m."**

Line 295 Here, one would guess that using inversion optimized fluxes, even made with another transport model would give a better match with observations than the unoptimized models and inventories used in the study (as mentioned on Line 510).

**Unfortunately, without a proper reference the text it is hard to surmise what is being referred to.**

Line 503 Said: "Modeled North Slope CH4 is underestimated throughout the measurement period

pointing to deficiencies in the wetland flux specifications over this ecoregion". What is the confidence level for this conclusion, given that the impact of the fluxes from outside of the study domain is considered significant, as discussed in Section 3.1.11?

**Since there is no section 3.1.11, we assume that the section being referred to is 3.3.5 which refers to the fact that a large part of the CO2 gradients observed are formed outside of the Arctic region. Couple of points to made in defense of this statement. First, there is a bunch evidence to suggest that CH4 emissions on the North Slope generated locally (Zhang et al., 2014; Sweeney et al., 2016; Hartley et al. and Floerchinger et al., 2019). Second, it is important point out that this offset is most notable over the North Slope which draws most of its background air for the Arctic Ocean where there we would expect minimal influence to the gradient.**

Line 508 Authors cite the improvements since the survey by Fisher et al (2014). This looks like a good point, should it be supported by a table or figure in the analysis?

**Fisher et al. was a broad survey of many models and demonstrates a complete lack of agreement between many forward models. As stated, in the text the next step is compare multiple forward models using the AIE bulk quantity as a benchmark to test the fidelity of these different models.**

Technical corrections

Line 141 fix square to round brackets
**Could not locate**
Line 173 what does sw mean in pco2sw? – need description
Done
Line 190 Table 1. In second LPJ-wsl reference – year is missing
Done
Line 264 words 'CH4 enhancements' can be removed - appear twice in a sentence
Not able to locate redundant use of CH4 enhancements